# ReEvo: Large Language Models as Hyper-Heuristics with Reflective Evolution

**Haoran Ye[1], Jiarui Wang[2], Zhiguang Cao[3], Federico Berto[4],**
**Chuanbo Hua[4], Haeyeon Kim[4], Jinkyoo Park[4], Guojie Song[1 5]** *

[1]National Key Laboratory of General Artificial Intelligence,
School of Intelligence Science and Technology, Peking University
[2]Southeast University  [3]Singapore Management University
[4]KAIST  [5]PKU-Wuhan Institute for Artificial Intelligence  AI4CO[†]

hrye@stu.pku.edu.cn, jiarui_wang@seu.edu.cn, zgcao@smu.edu.sg
{fberto, cbhua, haeyeonkim, jinkyoo.park}@kaist.ac.kr, gjsong@pku.edu.cn

Project Website: https://ai4co.github.io/reevo

## Abstract

The omnipresence of NP-hard combinatorial optimization problems (COPs) compels domain experts to engage in trial-and-error heuristic design. The long-standing endeavor of design automation has gained new momentum with the rise of large language models (LLMs). This paper introduces Language Hyper-Heuristics (LHHs), an emerging variant of Hyper-Heuristics that leverages LLMs for heuristic generation, featuring minimal manual intervention and open-ended heuristic spaces. To empower LHHs, we present Reflective Evolution (ReEvo), a novel integration of evolutionary search for efficiently exploring the heuristic space, and LLM reflections to provide verbal gradients within the space. Across five heterogeneous algorithmic types, six different COPs, and both white-box and black-box views of COPs, ReEvo yields state-of-the-art and competitive meta-heuristics, evolutionary algorithms, heuristics, and neural solvers, while being more sample-efficient than prior LHHs.

## 1 Introduction

NP-hard combinatorial optimization problems (COPs) pervade numerous real-world systems, each characterized by distinct constraints and objectives. The intrinsic complexity and heterogeneity of these problems compel domain experts to laboriously develop heuristics for their approximate solutions [23]. Automation of heuristic designs represents a longstanding pursuit.

Classic Hyper-Heuristics (HHs) automate heuristic design by searching for the best heuristic (combination) from a set of heuristics or heuristic components [64]. Despite decades of development, HHs are limited by heuristic spaces predefined by human experts [64]. The rise of large language models (LLMs) opens up new possibilities for HHs. This paper introduces the general concept of *Language Hyper-Heuristics (LHH)* to advance beyond preliminary attempts in individual COP settings [68, 46]. LHH constitutes an emerging variant of HH that utilizes LLMs for heuristic generations. It features minimal human intervention and open-ended heuristic spaces, showing promise to comprehensively shift the HH research paradigm.

---

*Correspondence to: Guojie Song <gjsong@pku.edu.cn>.
†Work made with contributions from the AI4CO open research community.

Pure LHH (e.g., LLM generations alone) is sample-inefficient and exhibits limited inference capability for black-box COPs. This work elicits the power of LHH with *Reflective Evolution* (`ReEvo`). ReEvo couples evolutionary search for efficiently exploring heuristic spaces, with self-reflections to boost the reasoning capabilities of LLMs. It emulates human experts by reflecting on the relative performance of two heuristics and gathering insights across iterations. This reflection approach is analogous to interpreting genetic cues and providing "*verbal gradient*" within search spaces. We introduce fitness landscape analysis and black-box prompting for reliable evaluation of LHHs. The dual-level reflections are shown to enhance heuristic search and induce verbal inference for black-box COPs, enabling `ReEvo` to outperform prior state-of-the-art (SOTA) LHH [47].

We introduce novel applications of LHHs and yield SOTA solvers with `ReEvo`: (1) We evolve penalty heuristics for Guided Local Search (GLS), which outperforms SOTA learning-based [52, 24, 75] and knowledge-based [1] (G)LS solvers. (2) We enhance Ant Colony Optimization (ACO) by evolving its heuristic measures, surpassing both neural-enhanced heuristics [94] and expert-designed heuristics [71, 6, 72, 17, 39]. (3) We refine the genetic algorithm (GA) for Electronic Design Automation (EDA) by evolving genetic operators, outperforming expert-designed GA [63] and the SOTA neural solver [31] for the Decap Placement Problem (DPP). (4) Compared to a classic HH [15], `ReEvo` generates superior constructive heuristics for the Traveling Salesman Problem (TSP). (5) We enhance the generalization of SOTA neural combinatorial optimization (NCO) solvers [37, 51] by evolving heuristics for attention reshaping. For example, we improve the optimality gap of POMO [37] from 52% to 29% and LEHD [51] from 3.2% to 3.0% on TSP1000, with negligible additional time overhead and no need for tuning neural models.

We summarize our contributions as follows. (1) We propose the concept of Language Hyper-Heuristics (LHHs), which bridges emerging attempts using LLMs for heuristic generation with a methodological group that enjoys decades of development. (2) We present Reflective Evolution (`ReEvo`), coupling evolutionary computation with humanoid reflections to elicit the power of LHHs. We introduce fitness landscape analysis and black-box prompting for reliable LHH evaluations, where `ReEvo` achieves SOTA sample efficiency. (3) We introduce novel applications of LHHs and present SOTA COP solvers with `ReEvo`, across five heterogeneous algorithmic types and six different COPs.

## 2 Related work

**Traditional Hyper-Heuristics.** Traditional HHs select the best performing heuristic from a pre-defined set [13] or generate new heuristics through the combination of simpler heuristic components [15, 104]. HHs offer a higher level of generality in solving various optimization problems [109, 96, 19, 44, 103, 58], but are limited by the heuristic space predefined by human experts.

**Neural Combinatorial Optimization.** Recent advances of NCO show promise in learning end-to-end solutions for COPs [2, 93, 3]. NCO can be regarded as a variant of HH, wherein neural architectures and solution pipelines define a heuristic space, and training algorithms search within it. A well-trained neural network (NN), under certain solution pipelines, represents a distinct heuristic. From this perspective, recent advancements in NCO HHs have led to better-aligned neural architectures [28, 51, 34, 73] and advanced solution pipelines [32, 52, 42, 89, 95, 12, 5] to define effective heuristic spaces, and improved training algorithms to efficiently explore heuristic spaces [33, 27, 14, 76, 18, 90, 79, 35], while targeting increasingly broader applications [9, 107, 54, 77]. In this work, we show that `ReEvo`-generated heuristics can outperform or enhance NCO methods.

**LLMs for code generation and optimization.** The rise of LLMs introduces new prospects for diverse fields [88, 82, 105, 25, 50, 99]. Among others, code generation capabilities of LLMs are utilized for code debugging [10, 49], enhancing code performance [55], solving algorithmic competition challenges [41, 70], robotics [38, 43, 81], and general task solving [92, 102]. Interleaving LLM generations with evaluations [74] yields powerful methods for prompt optimization [108, 83, 20], reinforcement learning (RL) reward design [53], algorithmic (self-)improvement [98, 48, 45], neural architecture search [8], and general solution optimization [91, 4, 80], with many under evolutionary frameworks [57, 87, 21, 7, 40]. Most related to `ReEvo`, concurrent efforts by Liu et al. [47] and Romera-Paredes et al. [68] leverage LLMs to develop heuristics for COPs. We go beyond and propose generic LHH for COPs, along with better sample efficiency, broader applications, more reliable evaluations, and improved heuristics. In addition, `ReEvo` contributes to a smoother fitness landscape,

showing the potential to enhance other tasks involving LLMs for optimization. We present further discussions in Appendix A.

**Self-reflections of LLMs.** Shinn et al. [70] propose to reinforce language agents via linguistic feedback, which is subsequently harnessed for various tasks [56, 84]. While Shinn et al. [70] leverage binary rewards indicating passing or failing test cases in programming, ReEvo extends the scope of verbal RL feedback to comparative analysis of two heuristics, analogous to verbal gradient information [66] within heuristic spaces. Also, ReEvo incorporates reflection within an evolutionary framework, presenting a novel and powerful integration.

## 3 Language Hyper-Heuristics for Combinatorial Optimization

HHs explore a search space of heuristic configurations to select or generate effective heuristics, indirectly optimizing the underlying COP. This dual-level framework is formally defined as follows.

**Definition 3.1** (Hyper-Heuristic). For COP with solution space $S$ and objective function $f : S \to \mathbb{R}$, a Hyper-Heuristic (HH) searches for the optimal heuristic $h^*$ in a heuristic space $H$ such that a meta-objective function $F : H \to \mathbb{R}$ is minimized, i.e., $h^* = \operatorname{argmin}_{h \in H} F(h)$.

Depending on how the heuristic space $H$ is defined, traditional HHs can be categorized into selection and generation HHs, both entailing manually defined heuristic primitives. Here, we introduce a novel variant of HHs, Language Hyper-Heuristics (LHH), wherein heuristics in $H$ are generated by LLMs. LHHs dispense with the need for predefined $H$, and instead leverage LLMs to explore an open-ended heuristic space. We recursively define LHHs as follows.

**Definition 3.2** (Language Hyper-Heuristic). A Language Hyper-Heuristic (LHH) is an HH variant where heuristics in $H$ are generated by LLMs.

In this work, we define the meta-objective function $F$ as the expected performance of a heuristic $h$ for certain COP. It is estimated by the average performance on a dataset of problem instances.

## 4 Language Hyper-Heuristic with ReEvo

LHH takes COP specifications as input and outputs the best inductive heuristic found for this COP. Vanilla LHH can be repeated LLM generations to randomly search the heuristic space, which is sample-inefficient and lacks reasoning capabilities for complex and black-box problems (see § 6). Therefore, we propose Reflective Evolution (ReEvo) to interpret genetic cues of evolutionary search and unleash the power of LHHs.

ReEvo is schematically illustrated in Fig. 1. Under an evolutionary framework, LLMs assume two roles: a *generator LLM* for generating individuals and a *reflector LLM* for guiding the generation with reflections. ReEvo, as an LHH, features a distinct individual encoding, where each individual is the code snippet of a heuristic. Its evolution begins with population initialization, followed by five iterative steps: selection, short-term reflection, crossover, long-term reflection, and elitist mutation. We evaluate the meta-objective of all heuristics, both after crossover and mutation. Our prompts are gathered in Appendix B.

**Individual encoding.** ReEvo optimizes toward best-performing heuristics via an evolutionary process, specifically a Genetic Programming (GP). It diverges from traditional GPs in that (1) individuals are code snippets generated by LLMs, and (2) individuals are not constrained by any predefined encoding format, except for adhering to a specified function signature.

**Population initialization.** ReEvo initializes a heuristic population by prompting the generator LLM with a task specification. A task specification contains COP descriptions (if available), heuristic designation, and heuristic functionality. Optionally, including seed heuristics, either trivial or expertly crafted to improve upon, can provide in-context examples that encourage valid heuristic generation and bias the search toward more promising directions.

A ReEvo iteration contains the following five sequential steps.

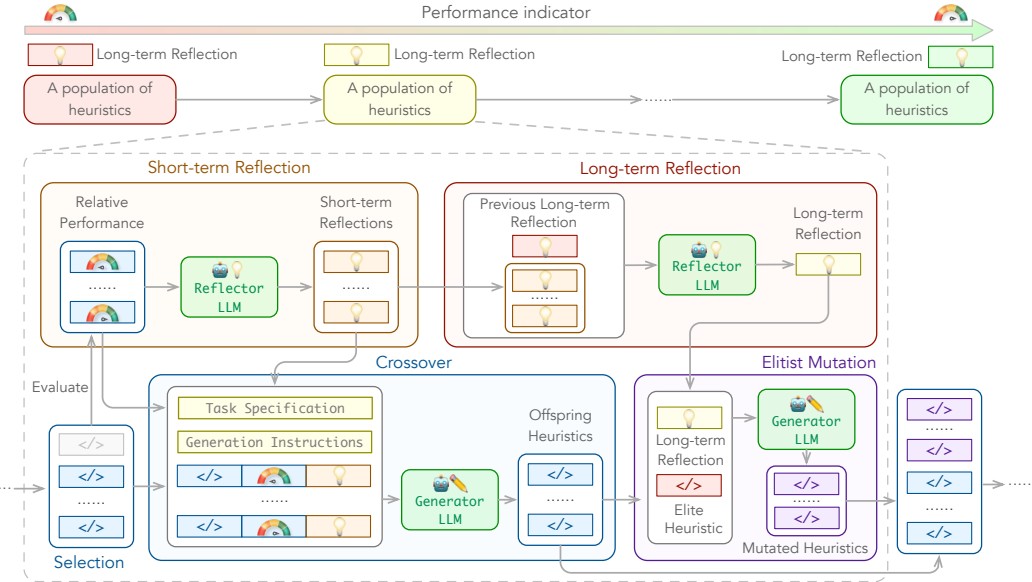

(a) ReEvo pipeline. **Top**: ReEvo evolves a population of heuristics. Insights and knowledge are verbalized as long-term reflections and accumulated throughout iterations. **Bottom**: A ReEvo iteration contains five sequential steps: selection, short-term reflection, crossover, long-term reflection, and elitist mutation.

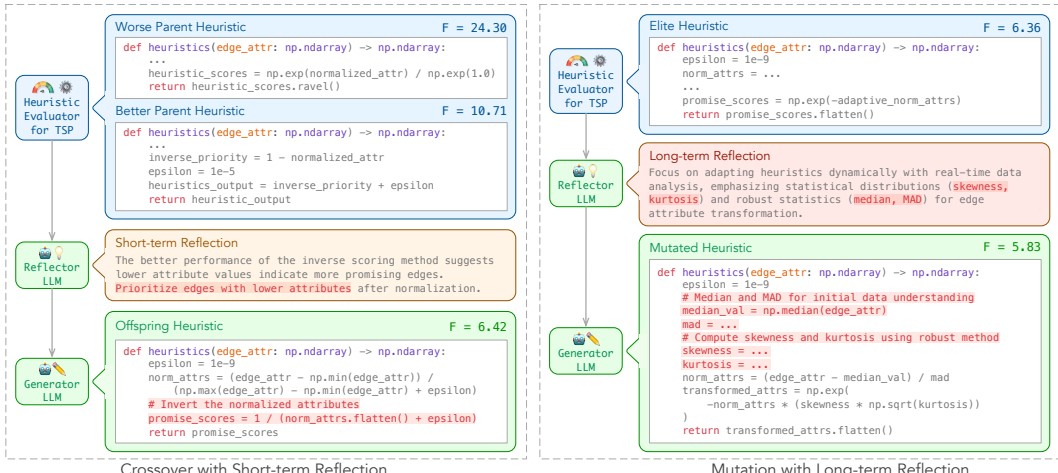

(b) Examples of reflections for black-box TSP. Heuristics are designed for Ant Colony Optimization (see § 5.2). **Left**: Given a pair parent heuristics, ReEvo correctly infers the TSP objective and generates a better offspring accordingly. **Right**: Given the elite heuristic and accumulated long-term reflections, ReEvo incorporates the suggested statistics and yields a better mutated heuristic.

Figure 1: An illustration of ReEvo.

**Selection.** ReEvo selects parent pairs from successfully executed heuristics at random, while avoiding pairing heuristics with an identical meta-objective value $F$.

**Short-term reflection.** For each pair of heuristic parents, the reflector LLM reflects upon their relative performance and gives hints accordingly for improved design. Unlike prior work [70], ReEvo integrates the reflections into evolutionary search and reflects by performing comparative analyses. Our proposed approach is analogous to interpreting genetic cues and providing verbal gradients within search spaces, which leads to smoother fitness landscapes and better search results (see § 6.1).

**Crossover.** ReEvo prompts the generator LLM to generate an offspring heuristic, given task specifications, a pair of parent heuristics, explicit indications of their relative performance, short-term reflections over the pair, and generation instructions.

**Long-term reflection.**    `ReEvo` accumulates expertise in improving heuristics via long-term reflections. The reflector LLM, given previous long-term reflections and newly gained short-term ones, summarizes them and gives hints for improved heuristic design.

**Elitist mutation.**    `ReEvo` employs an elitist mutation approach. Based on long-term reflections, the generator LLM samples multiple heuristics to improve the current best one. A mutation prompt consists of task specifications, the elite heuristic, long-term reflections, and generation instructions.

Viewing `ReEvo` from the perspective of an LLM agentic architecture [88], short-term reflections interpret the environmental feedback from each round of interaction. Long-term reflections distill accumulated experiences and knowledge, enabling them to be loaded into the inference context without causing memory blowups.

## 5    Heuristic generation with `ReEvo`

This section presents novel applications of LHH across heterogeneous algorithmic types and diverse COPs. With `ReEvo`, we yield state-of-the-art and competitive meta-heuristics, evolutionary algorithms, heuristics, and neural solvers.

Hyperparameters of `ReEvo` and detailed experimental setup are given in Appendix C. We apply `ReEvo` to different algorithmic types across six diverse COPs representative of different areas: Traveling Salesman Problem (TSP), Capacitated Vehicle Routing Problem (CVRP), and Orienteering Problem (OP) for routing problems; Multiple Knapsack Problem (MKP) for subset problems; Bin Packing Problem (BPP) for grouping problems; and Decap Placement Problem (DPP) for electronic design automation (EDA) problems. Details of the benchmark COPs are given in Appendix D. The best `ReEvo`-generated heuristics are collected in Appendix E.

### 5.1    Penalty heuristics for Guided Local Search

We evolve penalty heuristics for Guided Local Search (GLS) [1]. GLS interleaves local search with solution perturbation. The perturbation is guided by the penalty heuristics to maximize its utility. `ReEvo` searches for the penalty heuristic that leads to the best GLS performance.

We implement the best heuristic generated by `ReEvo` within KGLS [1] and refer to such coupling as KGLS-`ReEvo`. In Table 1, we compare KGLS-`ReEvo` with the original KGLS, other GLS variants [24, 75, 47], and SOTA NCO method that learns to improve a solution [52]. The results show that `ReEvo` can improve KGLS and outperform SOTA baselines. In addition, we use a single heuristic for TSP20 to 200, while NCO baselines require training models specific to each problem size.

Table 1: Evaluation results of different local search (LS) variants. We report optimality gaps and per-instance execution time.

| Method | Type | TSP20 | | TSP50 | | TSP100 | | TSP200 | |
|---|---|---|---|---|---|---|---|---|---|
| | | Opt. gap (%) | Time (s) | Opt. gap (%) | Time (s) | Opt. gap (%) | Time (s) | Opt. gap (%) | Time (s) |
| NeuOpt* [52] | LS+RL | 0.000 | 0.124 | 0.000 | 1.32 | 0.027 | 2.67 | 0.403 | 4.81 |
| GNNGLS [24] | GLS+SL | 0.000 | 0.116 | 0.052 | 3.83 | 0.705 | 6.78 | 3.522 | 9.92 |
| NeuralGLS† [75] | GLS+SL | 0.000 | 10.005 | 0.003 | 10.01 | 0.470 | 10.02 | 3.622 | 10.12 |
| EoH [47] | GLS+LHH | 0.000 | 0.563 | 0.000 | 1.90 | 0.025 | 5.87 | 0.338 | 17.52 |
| KGLS‡ [1] | GLS | 0.004 | 0.001 | 0.017 | 0.03 | 0.002 | 1.55 | 0.284 | 2.52 |
| KGLS-`ReEvo`‡ | GLS+LHH | **0.000** | **0.001** | **0.000** | **0.03** | **0.000** | **1.55** | **0.216** | **2.52** |

*: All instances are solved in one batch. D2A=1; T=500, 4000, 5000, and 5000 for 4 problem sizes, respectively.
†: The results are drawn from the original literature. ‡: They are based on our own GLS implementation.

### 5.2    Heuristic measures for Ant Colony Optimization

Solutions to COPs can be stochastically sampled, with heuristic measures indicating the promise of solution components and biasing the sampling. Ant Colony Optimization (ACO), which interleaves stochastic solution sampling with pheromone update, builds on this idea. We generate such heuristic measures for five different COPs: TSP, CVRP, OP, MKP, and BPP.

Under the ACO framework, we evaluate the best `ReEvo`-generated heuristics against the expert-designed ones and neural heuristics specifically learned for ACO [94]. The evolution curves displayed

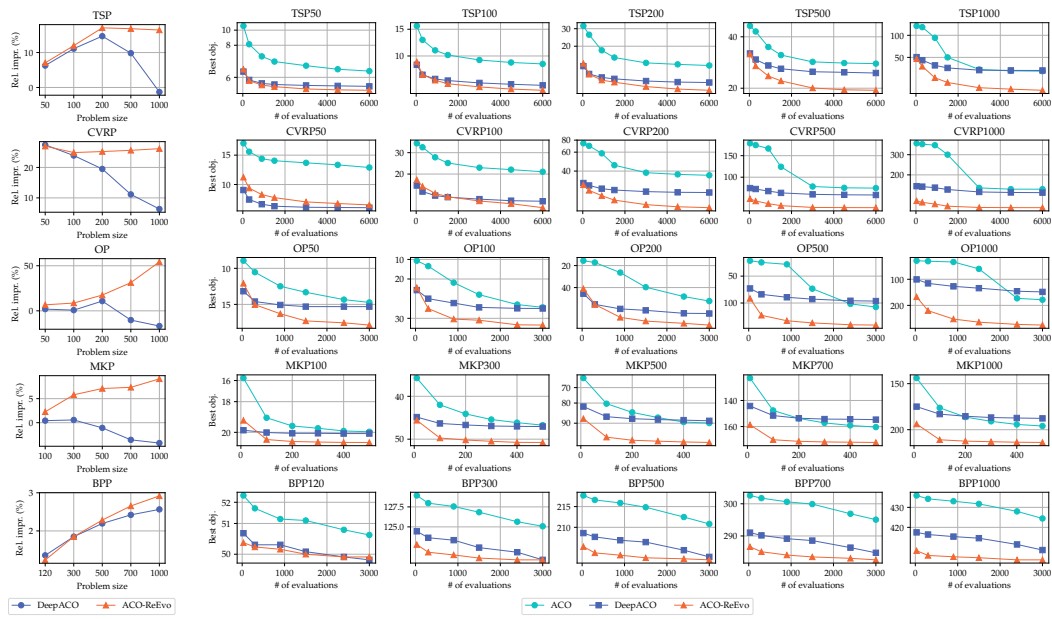

Figure 2: Comparative evaluations of ACO using expert-designed heuristics [71, 6, 72, 17, 39], neural heuristics [94], and ReEvo heuristics. For each COP, the same neural heuristic or the ReEvo heuristic is applied across all problem sizes; both heuristics are trained exclusively on the smallest problem size among the five. **Left**: Relative performance improvement of DeepACO and ReEvo over human baselines w.r.t. problem sizes. **Right**: ACO evolution curves, plotting the all-time best objective value w.r.t. the number of solution evaluations. The curves are averaged over three runs in which only small variances are observed (e.g., ∼ 0.01 for TSP50).

in Fig. 2 verify the consistent superiority of ReEvo across COPs and problem sizes. Notably, on 3 out of 5 COPs, ReEvo outperforms DeepACO [94] even when the latter overfits the test problem size (TSP50, OP50, and MKP100). We observe that most ReEvo-generated heuristics show consistent performance across problem sizes and distributions. Hence, their advantages grow as the distributional shift increases for neural heuristics.

## 5.3 Genetic operators for Electronic Design Automation

Expert-designed GAs are widely adopted in EDA [69, 97, 11, 26]. Besides directly solving EDA problems, GA-generated solutions can be used to train amortized neural solvers [31]. Here, we show that ReEvo can improve the expert-designed GAs and outperform DevFormer [31], the SOTA solver for the DPP problem. We sequentially evolve with ReEvo the crossover and mutation operators for the GA expert-designed by Park et al. [63]. Fig. 3 compares online and offline learned methods, DevFormer, the original expert-designed GA, and the GA with ReEvo-generated operators, showing that the ReEvo-designed GA outperforms previous methods and, importantly, both the expert-designed GA and DevFormer.

## 5.4 Constructive heuristics for the Traveling Salesman Problem

Heuristics can be used for deterministic solution construction by sequentially assigning values to each decision variable. We evaluate the constructive heuristic for TSP generated by ReEvo on real-world benchmark instances from TSPLIB [67] in Table 2. ReEvo can generate better heuristics than GHPP [15], a classic HH based on GP.

## 5.5 Attention reshaping for Neural Combinatorial Optimization

Autoregressive NCO solvers suffer from limited scaling-up generalization [29], partially due to the dispersion of attention scores [85]. Wang et al. [85] design a distance-aware heuristic to reshape

Table 2: Comparisons of constructive heuristics designed by human, GHPP [15], and `ReEvo`. We report the average optimality gap of each instance, where the baseline results are drawn from [15] and the results of `ReEvo` are averaged over 3 runs with different starting nodes.

| Instance | Nearest Neighbour | GHPP [15] | ReEvo |
|---|---|---|---|
| ts225 | 16.8 | 7.7 | **6.6** |
| rat99 | 21.8 | 14.1 | **12.4** |
| rl1889 | 23.7 | 21.1 | **17.5** |
| u1817 | 22.2 | 21.2 | **16.6** |
| d1655 | 23.9 | 18.7 | **17.5** |
| bier127 | 23.3 | 15.6 | **10.8** |
| lin318 | 25.8 | **14.3** | 16.6 |
| eil51 | 32.0 | 10.2 | **6.5** |
| d493 | 24.0 | 15.6 | **13.4** |
| kroB100 | 26.3 | 14.1 | **12.2** |
| kroC100 | 25.8 | 16.2 | **15.9** |

| Instance | Nearest Neighbour | GHPP [15] | ReEvo |
|---|---|---|---|
| ch130 | 25.7 | 14.8 | **9.4** |
| pr299 | 31.4 | **18.2** | 20.6 |
| fl417 | 32.4 | 22.7 | **19.2** |
| d657 | 29.7 | 16.3 | **16.0** |
| kroA150 | 26.1 | 15.6 | **11.6** |
| fl1577 | 25.0 | 17.6 | **12.1** |
| u724 | 28.5 | **15.5** | 16.9 |
| pr264 | 17.9 | 24.0 | **16.8** |
| pr226 | 24.6 | **15.5** | 18.0 |
| pr439 | 27.4 | 21.4 | **19.3** |
| Avg. opt. gap | 25.4 | 16.7 | **14.6** |

the attention scores, which improves the generalization of NCO solvers without additional training. However, the expert-designed attention-reshaping can be suboptimal and does not generalize across neural models or problem distributions.

Here we show that `ReEvo` can automatically and efficiently tailor attention reshaping for specific neural models and problem distributions of interest. We apply attention reshaping designed by experts [85] and `ReEvo` to two distinct model architectures: POMO with heavy encoder and light decoder [37], and LEHD with light encoder and heavy decoder [51]. On TSP and CVRP, Table 3 compares the original NCO solvers [37, 51], those with expert-designed attention reshaping [85], and those with `ReEvo`-designed attention reshaping. The results reveal that the `ReEvo`-generated heuristics can improve the original models and outperform their expert-designed counterparts. Note that implementing `ReEvo`-generated attention reshaping takes negligible additional time; e.g., solving a CVRP1000 with LEHD takes 50.0 seconds with reshaping, compared to 49.8 seconds without.

# 6 Evaluating `ReEvo`

## 6.1 Fitness landscape analysis

The fitness landscape of a searching algorithm depicts the structure and characteristics of its search space $F : H \to \mathbb{R}$ [59]. This understanding is essential for designing effective HHs. Here we introduce this technique to LHHs and evaluate the impact of reflections on the fitness landscape.

Traditionally, the neighborhood of a solution is defined as a set of solutions that can be reached after a single move of a certain heuristic. However, LHHs feature a probabilistic nature and open-ended search space, and we redefine its neighborhood as follows.

**Definition 6.1** (Neighborhood). Let $LLM$ denote an LHH move, $x$ a specific prompt, and $h_c$ the current heuristic. Given $LLM$ and $x$, the neighborhood of $h_c$ is defined as a set $\mathcal{N}$, where each

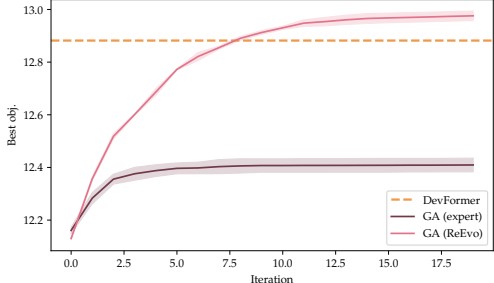

| Method | # of shots | Obj. ↑ |
|---|---|---|
| Pointer-PG [30] | 10,000 | $9.66_{\pm 0.206}$ |
| AM-PG [60] | 10,000 | $9.63_{\pm 0.587}$ |
| CNN-DQN [61] | 10,000 | $9.79_{\pm 0.267}$ |
| CNN-DDQN [100] | 10,000 | $9.63_{\pm 0.150}$ |
| Pointer-CRL [30] | Zero Shot | $9.59_{\pm 0.232}$ |
| AM-CRL [62] | Zero Shot | $9.56_{\pm 0.471}$ |
| DevFormer-CSE [31] | Zero Shot | $12.88_{\pm 0.003}$ |
| GA-expert [63] | 400 | $12.41_{\pm 0.026}$ |
| GA-ReEvo (ours) | 400 | $\mathbf{12.98}_{\pm \mathbf{0.018}}$ |

Figure 3: **Left**: Comparison of DevFormer [31], the expert-designed GA [63] and our `ReEvo`-designed GA on DPP. The evolution curves plot the best objective value over generations; the horizontal line indicates the reward of end-to-end solutions generated by DevFormer. **Right**: Evaluation results of DPP solvers. We report the number of solution generations and the average objective value of 100 test problems.

Table 3: Evaluation results for NCO solvers with and without different attention-reshaping heuristics.

| | Method | n = 200 | | n = 500 | | n = 1000 | |
|---|---|---|---|---|---|---|---|
| | | Obj. | Opt. gap (%) | Obj. | Opt. gap (%) | Obj. | Opt. gap (%) |
| TSP | POMO [37] | 11.16 | 4.40 | 22.21 | 34.43 | 35.19 | 52.11 |
| | POMO + DAR [85] | **11.12** | **3.98** | 21.63 | 30.95 | 33.32 | 44.05 |
| | POMO + ReEvo [75] | 11.12 | 4.02 | **20.54** | **24.32** | **29.86** | **29.08** |
| | LEHD [51] | 10.79 | 0.87 | 16.78 | 1.55 | 23.87 | 3.17 |
| | LEHD + DAR [85] | 10.79 | 0.89 | 16.79 | 1.62 | 23.87 | 3.19 |
| | LEHD + ReEvo | **10.77** | **0.74** | **16.78** | **1.55** | **23.82** | **2.97** |
| CVRP | POMO [37] | 22.39 | 10.93 | 50.12 | 33.76 | 145.40 | 289.48 |
| | POMO + DAR [85] | 22.36 | 10.78 | 50.23 | 34.05 | 144.24 | 286.37 |
| | POMO + ReEvo | **22.30** | **10.48** | **47.10** | **25.70** | **118.80** | **218.22** |
| | LEHD [51] | 20.92 | 3.68 | 38.61 | 3.03 | 39.12 | 4.79 |
| | LEHD + DAR [85] | 21.13 | 4.67 | 39.16 | 4.49 | 39.70 | 6.35 |
| | LEHD + ReEvo | **20.85** | **3.30** | **38.57** | **2.94** | **39.11** | **4.76** |

element $h \in \mathcal{N}$ represents a heuristic that $LLM$ can mutate $h_c$ into, in response to $x$:

$$\mathcal{N}(h_c) = \{h \mid LLM(h|h_c, x) > \xi\}. \tag{1}$$

Here, $LLM(h|h_c, x)$ denotes the probability of generating $h$ after prompting with $h_c$ and $x$, and $\xi$ is a small threshold value. In practice, the neighborhood can be approximated by sampling from the distribution $LLM(\cdot|h_c, x)$ for a large number of times.

We extend the concept of autocorrelation to LHHs under our definition of neighborhood. Autocorrelation reflects the ruggedness of a landscape, indicating the difficulty of a COP [59, 22].

**Definition 6.2** (Autocorrelation). Autocorrelation measures the correlation structure of a fitness landscape. It is derived from the autocorrelation function $r$ of a time series of fitness values, which are generated by a random walk on the landscape via neighboring points:

$$r_i = \frac{\sum_{t=1}^{T-i}(f_t - \bar{f})(f_{t+i} - \bar{f})}{\sum_{t=1}^{T}(f_t - \bar{f})^2}, \tag{2}$$

where $\bar{f}$ is the mean fitness of the points visited, $T$ is the size of the random walk, and $i$ is the time lag between points in the walk.

Based on the autocorrelation function, correlation length is defined below [86].

**Definition 6.3** (Correlation Length). Given an autocorrelation function $r$, the correlation length $l$ is formulated as $l = -1/\ln(|r_1|)$ for $r_1 \neq 0$. It reflects the ruggedness of a landscape, and smaller values indicate a more rugged landscape.

To perform autocorrelation analysis for ReEvo, we conduct random walks based on the neighborhood established with our crossover prompt either with or without short-term reflections. In practice, we set the population size to 1 and skip invalid heuristics; the selection always picks the current and last heuristics for short-term reflection and crossover, and we do not implement mutation.

Table 4 presents the correlation length and the average objective value of the random walks, where we generate ACO heuristics for TSP50. The correlation length is averaged over 3 runs each with 40 random walk steps, while the objective value is averaged over all $3 \times 40$ heuristics. The results verify that implementing reflection leads to a less rugged landscape and better search results. As discussed in § 4, reflections can function as verbal gradients that lead to better neighborhood structures.

Table 4: Autocorrelation analysis of ReEvo.

| | Correlation length ↑ | Objective ↓ |
|---|---|---|
| w/o reflection | 0.28 ± 0.07 | 12.08 ± 7.15 |
| w/ reflection | **1.28 ± 0.62** | **6.53 ± 0.60** |

## 6.2 Ablation studies

In this section, we investigate the effects of the proposed components of ReEvo with both white and *black-box* prompting.

**Black-box prompting.** We do not reveal any information related to the COPs and prompt LHHs in general forms (e.g., `edge_attr` in place of `distance_matrix`). Black-box settings allow reliable evaluations of LHHs in designing effective heuristics for novel and complex problems, rather than merely retrieving code tailored for prominent COPs from their parameterized knowledge.

We evaluate sampling LLM generations without evolution (LLM) and `ReEvo` without long-term reflections, short-term reflections, crossover, or mutation on generating ACO heuristics for TSP100. Table 5 shows that `ReEvo` enhances sample efficiency, and all its components positively contribute to its performance, both in white-box and black-box prompting.

Table 5: Ablation study of `ReEvo` components with both white and black-box prompting.

| Method | White-box ↓ | Black-box ↓ |
|---|---|---|
| LLM | $8.64 \pm 0.13$ | $9.74 \pm 0.54$ |
| w/o long-term reflections | $8.61 \pm 0.21$ | $9.32 \pm 0.71$ |
| w/o short-term reflections | $8.46 \pm 0.01$ | $9.05 \pm 0.83$ |
| w/o crossover | $8.45 \pm 0.02$ | $9.47 \pm 1.40$ |
| w/o mutation | $8.83 \pm 0.09$ | $9.34 \pm 0.96$ |
| `ReEvo` | $\mathbf{8.40} \pm 0.02$ | $\mathbf{8.96} \pm 0.82$ |

### 6.3 Comparative evaluations

This section compares `ReEvo` with EoH [47], a recent SOTA LHH that is more sample-efficient than FunSearch [68]. We adhere to the original code and (hyper)parameters of EoH. Our experiments apply both LHHs to generate ACO heuristics for TSP, CVRP, OP, MKP, and BPP, using black-box prompting and three LLMs: GPT-3.5 Turbo, GPT-4 Turbo, and Llama 3 (70B).

Fig. 4 compares EoH and `ReEvo`, and shows that `ReEvo` demonstrates superior sample efficiency. Besides the better neighborhood structure (§ 6.1), reflections facilitate explicit verbal inference of underlying black-box COP structures; we depict an example in Fig. 1 (b). The enhanced sample efficiency and inference capabilities of `ReEvo` are particularly useful for complex real-world problems, where the objective function is usually black-box and expensive to evaluate.

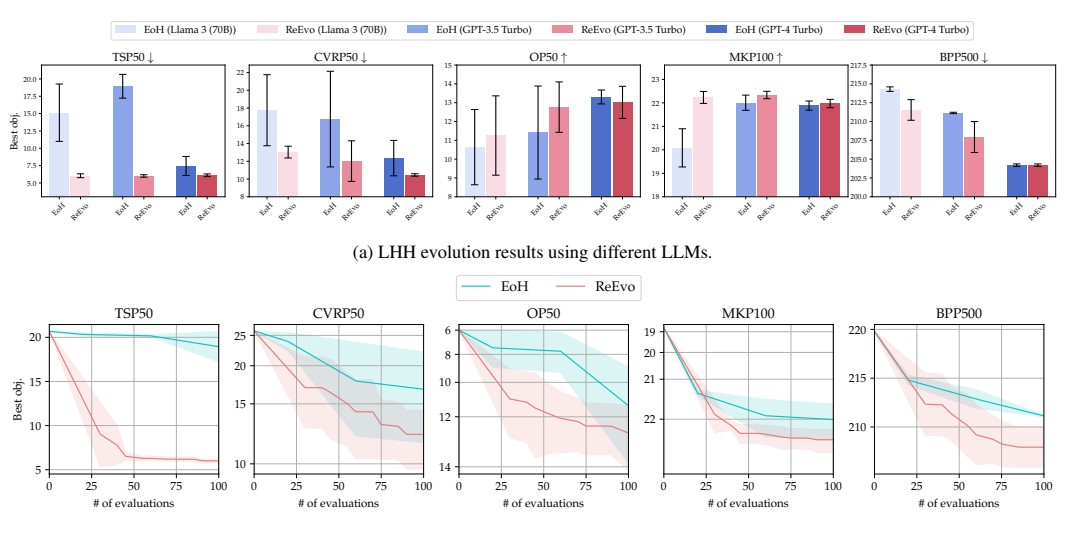

(a) LHH evolution results using different LLMs.

(b) LHH evolution curves using GPT-3.5 Turbo.

Figure 4: Comparisons between EoH [47] and `ReEvo` on five COPs with black-box prompting and using different LLMs. We perform three runs for each setting.

## 7 Discussions and limitations

**When to use `ReEvo` as an LHH.** Our experiments limit the number of heuristic evaluations to 100 shots and the results do not necessarily scale up. `ReEvo` is designed for scenarios where sample efficiency is crucial, such as real-world applications where heuristic evaluation can be costly. Allowing a large number of heuristic evaluations could obscure the impact of reflection or other prompting techniques, as reported by Zhang et al. [101].

**When to use `ReEvo` as an (alternative to) NCO/ML4CO method.** LHH can be categorized as an NCO/ML4CO method. However, to facilitate our discussion, we differentiate LHHs from

"traditional" NCO methods that usually train NN-parameterized heuristics via parameter adjustment. In § 5, we demonstrate that `ReEvo` can either outperform or enhance NCO methods. Below, we explore the complementary nature of LHH and NCO methods.

- **Rule-based v.s. NN-parameterized policies.** LHHs generate interpretable and rule-based heuristics (code snippets), while NCO generates black-box NN-parameterized policies. Interpretable heuristics offer insights for human designers and can be more reliable in practice when faced with dynamic environments, limited data, distributional shifts, or adversarial attacks. However, they may not be as expressive as neural networks and may underfit in complex environments.

- **Evolution and training.** LHHs require only less than 100 heuristic evaluations and about 5 minutes to evolve a strong heuristic, while many NCO methods usually require millions of samples and days of training. LHHs are more practical when solution evaluation is expensive.

- **Inference.** LHHs generate heuristics that are less demanding in terms of computational resources, as they do not require GPU during deployment. NCO methods require GPU for training and deployment, but they can also leverage the parallelism of GPU to potentially speed up inference.

- **Engineering efforts and inductive biases.** LHHs only need some text-based (and even black-box) explanations to guide the search. NCO requires the development of NN architectures, hyperparameters, and training strategies, where informed inductive biases and manual tuning are crucial to guarantee performance.

**The choice of LLMs for `ReEvo`.**   Reflection is more effective when using capable LLMs, such as GPT-3.5 Turbo and its successors, as discussed by Shinn et al. [70]. Currently, many open-source LLMs are not capable enough to guarantee statistically significant improvement of reflections [101]. However, as LLM capabilities improve, we only expect this paradigm to get better over time [70]. One can refer to [101] for extended evaluations based on more LLMs and problem settings.

**Benchmarking LHHs based on heuristic evaluations.**   We argue that benchmarking LHHs should prioritize the number of heuristic evaluations rather than LLM query budgets [101] due to the following reasons.

- Prioritizing scenarios where heuristic evaluations are costly leads to meaningful comparisons between LHHs. The performance of different LHH methods becomes nearly indistinguishable when a large number of heuristic evaluations are allowed [101].

- The overhead of LLM queries is negligible compared to real-world heuristic evaluations. LLM inference—whether via local models or commercial APIs—is highly cost-effective nowadays, with expenses averaging around $0.0003 per call in `ReEvo` using GPT-3.5-turbo, and response times of under one second on average for asynchronous API calls or batched inference. These costs are negligible compared to real-world heuristic evaluations, which, taking the toy EDA problem in this paper as an example, exceeds 20 minutes per evaluation.

- Benchmarking LHHs based on LLM inference costs presents additional challenges. Costs and processing time are driven by token usage rather than the number of queries, complicating the benchmarking process. For instance, EoH [47] requires heuristic descriptions before code generation, resulting in higher token usage. In contrast, although `ReEvo` involves more queries for reflections, it is more token-efficient when generating heuristics.

## 8  Conclusion

This paper presents Language Hyper-Heuristics (LHHs), a rising variant of HHs, alongside Reflective Evolution (`ReEvo`), an evolutionary framework to elicit the power of LHHs. Applying `ReEvo` across five heterogeneous algorithmic types, six different COPs, and both white-box and black-box views of COPs, we yield state-of-the-art and competitive meta-heuristics, evolutionary algorithms, heuristics, and neural solvers. Comparing against SOTA LHH [47], `ReEvo` demonstrates superior sample efficiency. The development of LHHs is still at its emerging stage. It is promising to explore their broader applications, better dual-level optimization architectures, and theoretical foundations. We also expect `ReEvo` to enrich the landscape of evolutionary computation, by showing that genetic cues can be interpreted and verbalized using LLMs.

## Acknowledgments and disclosure of funding

We are very grateful to Yuan Jiang, Yining Ma, Yifan Yang, AI4CO community, anonymous reviewers, and the area chair for valuable discussions and feedback. This work was supported by the National Natural Science Foundation of China (Grant No. 62276006); Wuhan East Lake High-Tech Development Zone National Comprehensive Experimental Base for Governance of Intelligent Society; the National Research Foundation, Singapore under its AI Singapore Programme (AISG Award No: AISG3-RP-2022-031); the National Research Foundation of Korea (NRF) grant funded by the Korea government (MSIT) (No. RS-2024-00410082); the Institute of Information & Communications Technology Planning & Evaluation (IITP)-Innovative Human Resource Development for Local Intellectualization program grant funded by the Korea government (MSIT) (IITP-2024-RS-2024-00436765).

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

# A Extended discussions

## A.1 Comparisons with EoH

Our work is developed concurrently with Evolution of Heuristics (EoH) [47], which establishes the groundwork for this emerging field. Nonetheless, our work extends the boundaries of LHH through three primary lenses: (1) the search algorithm, (2) the downstream CO applications, and (3) the evaluation methodologies.

- Search Algorithm: We introduce the Reflective Evolution, demonstrating its superior sample efficiency.
- Applications: Our work broadens the scope by applying LHH to five heterogeneous algorithmic types and six different COPs, advancing the state-of-the-art in GLS, EDA, ACO, and NCO.
- Evaluation Methodologies: We employ fitness landscape analysis to explore the underlying mechanisms of our proposed method; we establish black-box experimental settings to ensure reliable comparisons and practical relevance to real-world applications.

## A.2 Extended applications

ReEvo is generally applicable to other string-based optimization scenarios [57] as long as reflecting the relative performance of strings is meaningful. Preliminary experiments on prompt tuning verify the advantage of ReEvo over random search and vanilla genetic programming. Furthermore, we identify in reasoning-capable LLM approaches released after ReEvo such as OpenAI o1 [106] an interesting avenue of future works and experimentation that could yield even better sample efficiency and performance.

# B Prompts

We gather prompts used for ReEvo in this section. Our prompt structure is flexible and extensible. To adapt ReEvo to a new problem setting, one only needs to define its problem description, function description, and function signature.

## B.1 Common prompts

The prompt formats are given below. They are used for all COP settings.

> You are an expert in the domain of optimization heuristics. Your task is to design heuristics that can effectively solve optimization problems. Your response outputs Python code and nothing else. Format your code as a Python code string: "``` python ... ```".

Prompt 1: System prompt for generator LLM.

> You are an expert in the domain of optimization heuristics. Your task is to give hints to design better heuristics.

Prompt 2: System prompt for reflector LLM.

> Write a {function_name} function for {problem_description}
> {function_description}

Prompt 3: Task description.

> {task_description}
>
> {seed_function}
>
> Refer to the format of a trivial design above. Be very creative and give '{func_name}_v2'. Output code only and enclose your code with Python code block: ``` python ... ```.
>
> {initial_long−term_reflection}

Prompt 4: User prompt for population initialization.

```
Below are two {function_name} functions for {problem_description}
{ function_description }

You are provided with two code versions below, where the second version performs better than the first one.

[Worse code]
{worse_code}

[Better code]
{ better_code }

You respond with some hints for designing better heuristics, based on the two code versions and using less than 20 words.
```

Prompt 5: User prompt for short-term reflection.

The user prompt used for short-term reflection in black-box COPs is slightly different from the one used for white-box COPs. We explicitly ask the reflector LLM to infer the problem settings and to give hints about how the node and edge attributes correlate with the black-box objective value.

```
Below are two {function_name} functions for {problem_description}
{ function_description }

You are provided with two code versions below, where the second version performs better than the first one.

[Worse code]
{worse_code}

[Better code]
{ better_code }

Please infer the problem settings by comparing two code versions and give hints for designing better heuristics. You may give hints about how
        edge and node attributes correlate with the black−box objective value. Use less than 50 words.
```

Prompt 6: User prompt for short-term reflection on black-box COPs.

```
{ task_description }

[Worse code]
{ function_signature0 }
{worse_code}

[Better code]
{ function_signature1 }
{ better_code }

[ Reflection ]
{ short_term_reflection }

[Improved code]
Please write an improved function '{function_name}_v2', according to the reflection. Output code only and enclose your code with Python code
        block: ''' python ... '''.
```

Prompt 7: User prompt for crossover.

The function signature variables here are used to adjust function names with their versions, which is similar to the design in [68]. For example, when designing "heuristics", the worse code is named "heuristics_v0" while the better code "heuristics_v1". In Prompt 9, the elitist code is named "heuristic_v1".

```
Below is your prior long−term reflection on designing heuristics for {problem_description}
{prior_long−term_reflection }

Below are some newly gained insights.
{new_short−term_reflections}

Write constructive hints for designing better heuristics, based on prior reflections and new insights and using less than 50 words.
```

Prompt 8: User prompt for long-term reflection.

```
{ task_description }

[Prior   reflection ]
{long− term_reflection }

[Code]
{ function_signature1 }
{ elitist_code }

[Improved code]
Please  write  a  mutated  function  '{function_name}_v2',  according  to  the   reflection .  Output code only and  enclose  your  code  with  Python  code
    block:  ''' python  ...  '''.
```

Prompt 9: User prompt for elitist mutation.

## B.2  Problem-specific prompt components

Problem-specific prompt components are given below.

- Problem descriptions of all COP settings are given in Table 6.
- The function descriptions of all COP settings are presented in Table 7.  The descriptions crafted for black-box settings avoid disclosing any information that could link to the original COP.
- The function signatures are gathered in Prompt 10.
- The seed functions are shown in Prompt 11. The seed function used for TSP_constructive is drawn from [46]. The seed functions used for black-box ACO settings are expert-designed heuristics [71, 6, 72, 17, 39], while those used for while-box ACO settings are trivial all-ones matrices.
- The initial long-term reflections for some COP settings are presented in Prompt 12, while are left empty for the others.

```python
# TSP_NCO
def heuristics(distance_matrix: torch.Tensor) -> torch.Tensor:

# CVRP_NCO
def heuristics(distance_matrix: torch.Tensor, demands: torch.Tensor) -> torch.Tensor:

# DPP_GA_crossover
def crossover(parents: np.ndarray, n_pop: int) -> np.ndarray:

# DPP_GA_mutation
def mutation(population: np.ndarray, probe: int, prohibit: np.ndarray, size: int=100) -> np.ndarray:

# TSP_GLS
def heuristics(distance_matrix: np.ndarray) -> np.ndarray:

# TSP_ACO
def heuristics(distance_matrix: np.ndarray) -> np.ndarray:

# CVRP_ACO
def heuristics(distance_matrix: np.ndarray, coordinates: np.ndarray, demands: np.ndarray, capacity: int) -> np.ndarray:

# OP_ACO
def heuristics(prize: np.ndarray, distance: np.ndarray, maxlen: float) -> np.ndarray:

# MKP_ACO
def heuristics(prize: np.ndarray, weight: np.ndarray) -> np.ndarray:

# BPP_ACO
def heuristics(demand: np.ndarray, capacity: int) -> np.ndarray:

# TSP_ACO (black-box)
def heuristics(edge_attr: np.ndarray) -> np.ndarray:

# CVRP_ACO (black-box)
def heuristics(edge_attr: np.ndarray, node_attr: np.ndarray) -> np.ndarray: # For simplicity, we omit 'coordinates' and '
    capacity' after using capacity to normalize demands, i.e. node_attr

# OP_ACO (black-box)
def heuristics(node_attr: np.ndarray, edge_attr: np.ndarray, node_constraint: float) -> np.ndarray:

# MKP_ACO (black-box)
def heuristics(item_attr1: np.ndarray, item_attr2: np.ndarray) -> np.ndarray:

# BPP_ACO (black-box)
def heuristics(node_attr: np.ndarray, node_constraint: int) -> np.ndarray:
```

```
# TSP_constructive
def select_next_node(current_node: int, destination_node: int, unvisited_nodes: set, distance_matrix: np.ndarray) -> int:
```

Prompt 10: Function signatures used in ReEvo.

```
# TSP_NCO
def heuristics(distance_matrix: torch.Tensor) -> torch.Tensor:
    distance_matrix[distance_matrix == 0] = 1e5
    K = 100
    # Compute top-k nearest neighbors (smallest distances)
    values, indices = torch.topk(distance_matrix, k=K, largest=False, dim=1)
    heu = -distance_matrix.clone()
    # Create a mask where topk indices are True and others are False
    topk_mask = torch.zeros_like(distance_matrix, dtype=torch.bool)
    topk_mask.scatter_(1, indices, True)
    # Apply -log(d_ij) only to the top-k elements
    heu[topk_mask] = -torch.log(distance_matrix[topk_mask])
    return heu

# CVRP_NCO
def heuristics(distance_matrix: torch.Tensor, demands: torch.Tensor) -> torch.Tensor:
    return torch.zeros_like(distance_matrix)

# DPP_GA_crossover
def crossover(parents: np.ndarray, n_pop: int) -> np.ndarray:
    n_parents, n_decap = parents.shape
    # Split genomes into two halves
    left_halves = parents[:, :n_decap // 2]
    right_halves = parents[:, n_decap // 2:]
    # Create parent pairs
    parents_idx = np.stack([np.random.choice(range(n_parents), 2, replace=False) for _ in range(n_pop)])
    parents_left = left_halves[parents_idx[:, 0]]
    parents_right = right_halves[parents_idx[:, 1]]
    # Create offspring
    offspring = np.concatenate([parents_left, parents_right], axis=1)
    return offspring

# DPP_GA_mutation
def mutation(population: np.ndarray, probe: int, prohibit: np.ndarray, size: int=100) -> np.ndarray:
    n_pop, n_decap = population.shape
    for i in range(n_pop):
        ind = population[i]
        unique_actions = np.unique(population[i])
        if len(unique_actions) < n_decap:
            # Find the indices wherein the action is taken the second time
            dup_idx = []
            action_set = set()
            for j, action in enumerate(ind):
                if action in action_set:
                    dup_idx.append(j)
                action_set.add(action)

            # Mutate the duplicated actions
            infeasible_actions = np.concatenate([prohibit, [probe], unique_actions])
            feasible_actions = np.setdiff1d(np.arange(size), infeasible_actions)
            assert n_decap - len(unique_actions) == len(dup_idx)
            new_actions = np.random.choice(feasible_actions, len(dup_idx), replace=False)
            ind[dup_idx] = new_actions
    return population

# TSP_GLS
def heuristics(distance_matrix: np.ndarray) -> np.ndarray:
    return distance_matrix

# TSP_ACO
def heuristics(distance_matrix: np.ndarray) -> np.ndarray:
    return 1 / distance_matrix

# CVRP_ACO
def heuristics(distance_matrix: np.ndarray, coordinates: np.ndarray, demands: np.ndarray, capacity: int) -> np.ndarray:
    return 1 / distance_matrix

# OP_ACO
def heuristics(prize: np.ndarray, distance: np.ndarray, maxlen: float) -> np.ndarray:
    return prize[np.newaxis, :] / distance

# MKP_ACO
def heuristics(prize: np.ndarray, weight: np.ndarray) -> np.ndarray:
    return prize / np.sum(weight, axis=1)

# BPP_ACO
def heuristics(demand: np.ndarray, capacity: int) -> np.ndarray:
    return np.tile(demand/demand.max(), (demand.shape[0], 1))

# TSP_ACO (black-box)
def heuristics(edge_attr: np.ndarray) -> np.ndarray:
    return np.ones(edge_attr.shape[0])
```

```
# CVRP_ACO (black-box)
def heuristics(edge_attr: np.ndarray, node_attr: np.ndarray) -> np.ndarray:
    return np.ones_like(edge_attr)

# OP_ACO (black-box)
def heuristics(node_attr: np.ndarray, edge_attr: np.ndarray, edge_constraint: float) -> np.ndarray:
    return np.ones_like(edge_attr)

# MKP_ACO (black-box)
def heuristics(item_attr1: np.ndarray, item_attr2: np.ndarray) -> np.ndarray:
    n, m = item_attr2.shape
    return np.ones(n,)

# BPP_ACO (black-box)
def heuristics(node_attr: np.ndarray, node_constraint: int) -> np.ndarray:
    n = node_attr.shape[0]
    return np.ones((n, n))

# TSP_constructive
def select_next_node(current_node: int, destination_node: int, unvisited_nodes: set, distance_matrix: np.ndarray) -> int:
    threshold = 0.7
    c1, c2, c3, c4 = 0.4, 0.3, 0.2, 0.1
    scores = {}
    for node in unvisited_nodes:
        all_distances = [distance_matrix[node][i] for i in unvisited_nodes if i != node]
        average_distance_to_unvisited = np.mean(all_distances)
        std_dev_distance_to_unvisited = np.std(all_distances)
        score = c1 * distance_matrix[current_node][node] - c2 * average_distance_to_unvisited + c3 *
      std_dev_distance_to_unvisited - c4 * distance_matrix[destination_node][node]
        scores[node] = score
    next_node = min(scores, key=scores.get)
    return next_node
```

Prompt 11: Seed heuristics used for `ReEvo`.

```
# White−box COP_ACO
− Try combining various factors to determine how promising it is to select a solution component.
− Try sparsifying the matrix by setting unpromising elements to zero.

# TSP_constructive
− Try look−ahead mechanisms.
```

Prompt 12: Initial long-term reflections

Table 7: Function descriptions used in prompts.

| Problem | Function description |
|---|---|
| TSP_NCO | The 'heuristics' function takes as input a distance matrix and returns prior indicators of how bad it is to include each edge in a solution. The return is of the same shape as the input. The heuristics should contain negative values for undesirable edges and positive values for promising ones. Use efficient vectorized implementations. |
| CVRP_NCO | The 'heuristics' function takes as input a distance matrix (shape: n by n) and a vector of customer demands (shape: n), where the depot node is indexed by 0 and the customer demands are normalized by the total vehicle capacity. It returns prior indicators of how promising it is to include each edge in a solution. The return is of the same shape as the distance matrix. The heuristics should contain negative values for undesirable edges and positive values for promising ones. Use efficient vectorized implementations. |
| DPP_GA_crossover | The 'crossover' function takes as input a 2D NumPy array parents and an integer n_pop. The function performs a genetic crossover operation on parents to generate n_pop offspring. Use vectorized implementation if possible. |

| Problem | Function description |
|---|---|
| DPP_GA_mutation | The 'mutation' function modifies a given 2D population array to ensure exploration of the genetic algorithm. You may also take into account the feasibility of each individual. An individual is considered feasible if all its elements are unique and none are listed in the prohibited array or match the probe value. Use a vectorized implementation if possible.
The function takes as input the below arguments:
- population (np.ndarray): Population of individuals; shape: (P, n_decap).
- probe (int): Probe value; each element in the population should not be equal to this value.
- prohibit (np.ndarray): Prohibit values; each element in the population should not be in this set.
- size (int): Size of the PDN; each element in the population should be in the range [0, size). |
| TSP_GLS | The 'heuristics' function takes as input a distance matrix, and returns prior indicators of how bad it is to include each edge in a solution. The return is of the same shape as the input. |
| TSP_ACO | The 'heuristics' function takes as input a distance matrix, and returns prior indicators of how promising it is to include each edge in a solution. The return is of the same shape as the input. |
| TSP_ACO_black-box | The 'heuristics' function takes as input a matrix of edge attributes with shape '(n_edges, n_attributes)', where 'n_attributes=1' in this case. It computes prior indicators of how promising it is to include each edge in a solution. The return is of the shape of '(n_edges,)'. |
| CVRP_ACO | The 'heuristics' function takes as input a distance matrix (shape: n by n), Euclidean coordinates of nodes (shape: n by 2), a vector of customer demands (shape: n), and the integer capacity of vehicle capacity. It returns prior indicators of how promising it is to include each edge in a solution. The return is of the same shape as the distance matrix. The depot node is indexed by 0. |
| CVRP_ACO_black-box | The 'heuristics' function takes as input a matrix of edge attributes (shape: n by n) and a vector of node attributes (shape: n). A special node is indexed by 0. 'heuristics' returns prior indicators of how promising it is to include each edge in a solution. The return is of the same shape as the input matrix of edge attributes. |
| OP_ACO | Suppose 'n' represents the number of nodes in the problem, with the depot being the first node. The 'heuristics' function takes as input a 'prize' array of shape (n,), a 'distance' matrix of shape (n,n), and a 'max_len' float which is the constraint to total travel distance, and it returns 'heuristics' of shape (n, n), where 'heuristics[i][j]' indicates the promise of including the edge from node #i to node #j in the solution. |
| OP_ACO_black-box | The 'heuristics' function takes as input a vector of node attributes (shape: n), a matrix of edge attributes (shape: n by n), and a constraint imposed on the sum of edge attributes. A special node is indexed by 0. 'heuristics' returns prior indicators of how promising it is to include each edge in a solution. The return is of the same shape as the input matrix of edge attributes. |
| MKP_ACO | Suppose 'n' indicates the scale of the problem, and 'm' is the dimension of weights each item has. The constraint of each dimension is fixed to 1. The 'heuristics' function takes as input a 'prize' of shape (n,), a 'weight' of shape (n, m), and returns 'heuristics' of shape (n,). 'heuristics[i]' indicates how promising it is to include item i in the solution. |
| MKP_ACO_black-box | Suppose 'n' indicates the scale of the problem, and 'm' is the dimension of some attributes each involved item has. The 'heuristics' function takes as input an 'item_attr1' of shape (n,), an 'item_attr2' of shape (n, m), and returns 'heuristics' of shape (n,). 'heuristics[i]' indicates how promising it is to include item i in the solution. |

| Problem | Function description |
|---|---|
| BPP_ACO | Suppose 'n' represents the number of items in the problem. The heuristics function takes as input a 'demand' array of shape (n,) and an integer as the capacity of every bin, and it returns a 'heuristics' array of shape (n,n). 'heuristics[i][j]' indicates how promising it is to put item i and item j in the same bin. |
| BPP_ACO_black-box | Suppose 'n' represents the scale of the problem. The heuristics function takes as input an 'item_attr' array of shape (n,) and an integer as a certain constraint imposed on the item attributes. The heuristics function returns a 'heuristics' array of shape (n, n). 'heuristics[i][j]' indicates how promising it is to group item i and item j. |
| TSP_constructive | The select_next_node function takes as input the current node, the destination node, a set of unvisited nodes, and a distance matrix, and returns the next node to visit. |

# C   Detailed experimental setup

**Hyperparameters for `ReEvo`.**   Unless otherwise stated, we adopt the parameters in Table 8 for `ReEvo` runs. During initialization, the LLM temperature is added by 0.3 to diversify the initial population.

**Heuristic generation pipeline.**   We perform 3 `ReEvo` runs for each COP setting. Unless otherwise stated, the heuristic with the best validation performance is selected for final testing on 64 held-out instances.

**Cost and hardware.**   When the hardware permits, heuristics from the same generation are generated, reflected upon, and evaluated in parallel. The duration of a single `ReEvo` run can range from approximately two minutes to hours, depending on the evaluation runtime and the hardware used. Each run costs about $0.06 when using GPT3.5 Turbo. When conducting runtime comparisons, we employ a single core of an AMD EPYC 7742 CPU and an NVIDIA GeForce RTX 3090 GPU.

## C.1   Penalty heuristics for Guided Local Search

Guided Local Search (GLS) explores solution space through local search operations under the guidance of heuristics. We aim to use `ReEvo` to find the most effective heuristics to enhance GLS. In our experimental setup, we employed a variation of the classical GLS algorithm [78] that incorporated perturbation phases [1], wherein edges with higher heuristic values will be prioritized for penalization. In the training phase, we evaluate each heuristic with TSP200 using 1200 GLS iterations. For generating results in Table 1, we use the parameters in Table 9. The iterations stop when reaching the predefined threshold or when the optimality gap is reduced to zero.

## C.2   Heuristic measures for Ant Colony Optimization

Ant Colony Optimization is an evolutionary algorithm that interleaves solution samplings with the update of pheromone trails. Stochastic solution samplings are biased toward more promising solution space by heuristics, and `ReEvo` searches for the best of such heuristics. For more details, please refer to [94].

Table 10 presents the ACO parameters used for heuristic evaluations during LHH evolution. They are adjusted to maximize ACO performance while ensuring efficient evaluations. Instance generations and ACO implementations follow Ye et al. [94]. To conduct tests in Fig. 2, we increase the number of iterations to ensure full convergence.

## C.3   Genetic operators for Electronic Design Automation

Here we briefly introduce the expert-design GA for DPP. Further details can be found in [31, Appendix B].

The GA designed by Kim et al. [31] is utilized as an expert policy to collect expert guiding labels for imitation learning. The GA is a widely used search heuristic method for the Decoupling Capacitor Placement Problem (DPP), which aims to find the optimal placement of a given number of decoupling capacitors (decaps) on a Power Distribution Network (PDN) with a probing port and 0-15 keep-out regions to best suppress the impedance of the probing port.

Key aspects of the designed GA include:

Table 6: Problem descriptions used in prompts.

| Problem | Problem description |
|---------|---------------------|
| TSP_NCO | Assisting in solving the Traveling Salesman Problem (TSP) with some prior heuristics. TSP requires finding the shortest path that visits all given nodes and returns to the starting node. |
| CVRP_NCO | Assisting in solving Capacitated Vehicle Routing Problem (CVRP) with some prior heuristics. CVRP requires finding the shortest path that visits all given nodes and returns to the starting node. Each node has a demand and each vehicle has a capacity. The total demand of the nodes visited by a vehicle cannot exceed the vehicle capacity. When the total demand exceeds the vehicle capacity, the vehicle must return to the starting node. |
| DPP_GA | Assisting in solving black-box decap placement problem with genetic algorithm. The problem requires finding the optimal placement of decaps in a given power grid. |
| TSP_GLS | Solving Traveling Salesman Problem (TSP) via guided local search. TSP requires finding the shortest path that visits all given nodes and returns to the starting node. |
| TSP_ACO | Solving Traveling Salesman Problem (TSP) via stochastic solution sampling following "heuristics". TSP requires finding the shortest path that visits all given nodes and returns to the starting node. |
| TSP_ACO_black-box | Solving a black-box graph combinatorial optimization problem via stochastic solution sampling following "heuristics". |
| CVRP_ACO | Solving Capacitated Vehicle Routing Problem (CVRP) via stochastic solution sampling. CVRP requires finding the shortest path that visits all given nodes and returns to the starting node. Each node has a demand and each vehicle has a capacity. The total demand of the nodes visited by a vehicle cannot exceed the vehicle capacity. When the total demand exceeds the vehicle capacity, the vehicle must return to the starting node. |
| CVRP_ACO_black-box | Solving a black-box graph combinatorial optimization problem via stochastic solution sampling following "heuristics". |
| OP_ACO | Solving Orienteering Problem (OP) via stochastic solution sampling following "heuristics". OP is an optimization problem where the goal is to find the most rewarding route, starting from a depot, visiting a subset of nodes with associated prizes, and returning to the depot within a specified travel distance. |
| OP_ACO_black-box | Solving a black-box graph combinatorial optimization problem via stochastic solution sampling following "heuristics". |
| MKP_ACO | Solving Multiple Knapsack Problems (MKP) through stochastic solution sampling based on "heuristics". MKP involves selecting a subset of items to maximize the total prize collected, subject to multi-dimensional maximum weight constraints. |
| MKP_ACO_black-box | Solving a black-box combinatorial optimization problem via stochastic solution sampling following "heuristics". |
| BPP_ACO | Solving Bin Packing Problem (BPP). BPP requires packing a set of items of various sizes into the smallest number of fixed-sized bins. |
| BPP_ACO_black-box | Solving a black-box combinatorial optimization problem via stochastic solution sampling following "heuristics". |
| TSP_constructive | Solving Traveling Salesman Problem (TSP) with constructive heuristics. TSP requires finding the shortest path that visits all given nodes and returns to the starting node. |

Table 8: Parameters of `ReEvo`.

| Parameter | Value |
|---|---|
| LLM (generator and reflector) | gpt-3.5-turbo |
| LLM temperature (generator and reflector) | 1 |
| Population size | 10 |
| Number of initial generation | 30 |
| Maximum number of evaluations | 100 |
| Crossover rate | 1 |
| Mutation rate | 0.5 |

Table 9: GLS parameters used for the evaluations in Table 1.

| Problem | Perturbation moves | Number of iterations | Scale parameter $\lambda$ |
|---|---|---|---|
| TSP20 | 5 | 73 | |
| TSP50 | 30 | 175 | 0.1 |
| TSP100 | 40 | 1800 | |
| TSP200 | 40 | 800 | |

- **Encoding and initialization.** The GA generates an initial population randomly, and each solution consists of a set of numbers representing decap locations on the PDN. The population size is fixed to 20, and each solution is evaluated and sorted based on its objective value.

- **Elitism.** After the initial population is formulated, the top-performing solutions (elite population) are kept for the next generation. The size of the elite population is predefined as 4.

- **Selection.** The better half of the population is selected for crossover.

- **Crossover.** This process generates new population candidates by dividing each solution from the selected population in half and performing random crossover.

- **Mutation.** After crossover, solutions with overlapping numbers are replaced with random numbers while avoiding locations of the probing port and keep-out regions.

In this work, we sequentially optimize the crossover and mutation operators using `ReEvo`. When optimizing crossover, all other components of the GA pipeline remain identical to the expert-designed one. When optimizing mutation, we additionally set the crossover operator to the best one previously generated by `ReEvo`.

During training, we evaluate $F$ on three training instances randomly generated following [31, Appendix A.5]. The evaluation on each instance runs 10 GA iterations and returns the objective value of the best-performing solution. For the final test in Fig. 3, we utilize the same test dataset as in [31].

### C.4  Attention reshaping for Neural Combinatorial Optimization

For autoregressive NCO solvers, e.g. POMO [37] and LEHD [51], the last decoder layer outputs the logits of the next node to visit. Then, the attention-reshaping heuristic values are added to the logits before masking, logit clipping, and softmax operation.

For the autoregressive NCO models with a heavy encoder and a light decoder, the last decoder layer computes logits using [36]

$$
u_{(c)j} = \begin{cases} C \cdot \tanh\left(\dfrac{\mathbf{q}_{(c)}^T \mathbf{k}_j}{\sqrt{d_\mathrm{k}}}\right) & \text{if } j \neq \pi_{t'} \quad \forall t' < t \\ -\infty & \text{otherwise.} \end{cases} \tag{3}
$$

Table 10: ACO parameters used for heuristic evaluations during training.

| Problem | Population size | Number of iterations |
|---|---|---|
| TSP | 30 | 100 |
| CVRP | 30 | 100 |
| OP | 20 | 50 |
| MKP | 10 | 50 |
| BPP | 20 | 15 |

Here, $u_{(c)j}$ is the compatibility between current context and node $j$, $C$ a constant for logit clipping, $\mathbf{q}_{(c)}$ the query embedding of the current context, $\mathbf{k}_j$ the key embedding of node $j$, and $d_k$ the query/key dimensionality. For each node $j$ already visited, i.e. $j = \pi_{t'}, \exists t' < t$, $u_{(c)j}$ is masked.

We reshape the attention scores by using

$$u_{(c)j} = \begin{cases} C \cdot \tanh\left(\frac{\mathbf{q}_{(c)}^T \mathbf{k}_j + h_{(c)j}}{\sqrt{d_k}}\right) & \text{if } j \neq \pi_{t'} \quad \forall t' < t \\ -\infty & \text{otherwise.} \end{cases} \tag{4}$$

$h_{(c)j}$ is computed via attention-reshaping heuristics. In practice, for TSP and CVRP, $h_{(c)j} = \mathbf{H}_{c,j}$, where $\mathbf{H}$ is the heuristic matrix and $c$ is simplified to the current node.

For the autoregressive NCO models with a light encoder and a heavy decoder [51], or only a decoder [14], the last decoder layer computes logits using:

$$u_i = W_o \mathbf{h}_i, \tag{5}$$

where $W_o$ is a learnable matrix at the output layer and node $i$ is among the available nodes. We reshape the logits with

$$u_i = W_o \mathbf{h}_i + \mathbf{H}_{c,i}, \tag{6}$$

where node $c$ is the current node.

For evaluations in Table 3, we generalize the models trained on TSP100 and CVRP100 to larger instances with 200, 500, and 1000 nodes. For TSP, we apply the same `ReEvo`-generated heuristic across all sizes, whereas for the CVRP, we use distinct heuristics for each size due to the observed variations in desirable heuristics.

# D    Benchmark problems

## D.1    Traveling Salesman Problem

**Definition.**    The Traveling Salesman Problem (TSP) is a classic optimization challenge that seeks the shortest possible route for a salesman to visit each city in a list exactly once and return to the origin city.

**Instance generation.**    Nodes are sampled uniformly from $[0,1]^2$ unit for the synthetic datasets.

## D.2    Capacitated Vehicle Routing Problem

**Definition.**    The Capacitated Vehicle Routing Problem (CVRP) extends the TSP by adding constraints on vehicle capacity. Each vehicle can carry a limited load, and the objective is to minimize the total distance traveled while delivering goods to various locations.

**Instance generation.**    For § 5.2, We follow DeepACO [94]. Customer locations are sampled uniformly in the unit square; customer demands are sampled from the discrete set $\{1, 2, \ldots, 9\}$; the capacity of each vehicle is set to 50; the depot is located at the center of the unit square. For § 5.5, we use the test instances provided by LEHD [51].

## D.3    Orienteering Problem

**Definition.**    In the Orienteering Problem (OP), the goal is to maximize the total score collected by visiting nodes while subject to a maximum tour length constraint.

**Instance generation.**    The generation of synthetic datasets aligns with DeepACO [94]. We uniformly sample the nodes, including the depot node, from the unit $[0,1]^2$. We use a challenging prize distribution [36]: $p_i = (1 + \lfloor 99 \cdot \frac{d_{0i}}{\max_{j=1}^n d_{0j}} \rfloor)/100$, where $d_{0i}$ is the distance between the depot and node $i$. The maximum length constraint is also designed to be challenging. As suggested by Kool et al. [36], we set it to 3, 4, 5, 8, and 12 for OP50, OP100, OP200, OP500, and OP1000, respectively.

## D.4    Multiple Knapsack Problem

**Definition.**    The Multiple Knapsack Problem (MKP) involves distributing a set of items, each with a given weight and value, among multiple knapsacks to maximize the total value without exceeding the capacity of any knapsack.

**Instance generation.** Instance generation follows DeepACO [94]. The values and weights are uniformly sampled from [0, 1]. To make all instances well-stated, we uniformly sample $c_i$ from $(\max_j w_{ij}, \sum_j w_{ij})$.

## D.5 Bin Packing Problem

**Definition.** The Bin Packing Problem requires packing objects of different volumes into a finite number of bins or containers of a fixed volume in a way that minimizes the number of bins used. It is widely applicable in manufacturing, shipping, and storage optimization.

**Instance generation.** Following Levine and Ducatelle [39], we set the bin capacity to 150, and item sizes are uniformly sampled between 20 and 100.

## D.6 Decap Placement Problem

**Definition.** The Decap Placement Problem (DPP) is a critical hardware design optimization issue that involves finding the optimal placement of decoupling capacitors (decap) within a power distribution network (PDN) to enhance power integrity (PI) [65, 69, 16, 63]. Decoupling capacitors are hardware components that help reduce power noise and ensure a stable supply of power to the operating integrated circuits within hardware devices such as CPUs, GPUs, and AI accelerators [65]. The DPP is formulated as a black-box contextual optimization problem, where the goal is to determine the best positions for a set of decaps to maximize the PI objective. This objective is contextualized by the target hardware's feature vectors, with the constraint of a limited number of decaps. Interested readers can refer to [31] for more details.

**Instance generation.** $10 \times 10$ PDN instances are used. We generate training and validation instances following Kim et al. [31]. The test instances are directly drawn from [31].

# E Generated heuristics

This section presents the best heuristics generated by `ReEvo` for all problem settings.

```python
def heuristics(distance_matrix: torch.Tensor) -> torch.Tensor:
    distance_matrix[distance_matrix == 0] = 1e5
    beta = 0.3
    gamma = 0.5
    reciprocal = -1 / distance_matrix
    log_values = -torch.log(distance_matrix)
    local_heu = log_values + beta * (reciprocal.mean(dim=1, keepdim=True) - reciprocal)
    global_mean = distance_matrix.mean()
    global_heu = -gamma * torch.log(torch.abs(distance_matrix - global_mean))
    heu = local_heu + global_heu
    return heu
```

Heuristic 1: The best `ReEvo`-generated heuristic for TSP_NCO_POMO.

```python
def heuristics(distance_matrix: torch.Tensor) -> torch.Tensor:
    n = distance_matrix.size(0)
    # Calculate the average distance for each node with symmetrical adjustments
    avg_distances = (torch.sum(distance_matrix, dim=1, keepdim=True) + torch.sum(distance_matrix, dim=0, keepdim=True) - 2 *
        torch.diag(distance_matrix).unsqueeze(1)) / (2 * (n - 1))
    # Calculate heuristics based on the difference between each distance and the average distances with emphasis on node-
      centric averages
    heuristics = 2 * (distance_matrix - avg_distances) + 0.5 * (distance_matrix - torch.mean(distance_matrix, dim=1, keepdim
        =True))
    # Normalize the heuristics to have a mean of 0 and standard deviation of 1
    heuristics = (heuristics - torch.mean(heuristics)) / torch.std(heuristics)
    return heuristics
```

Heuristic 2: The best `ReEvo`-generated heuristic for TSP_NCO_LEHD.

```python
# For CVRP200
def heuristics(distance_matrix: torch.Tensor, demands: torch.Tensor) -> torch.Tensor:
    n = distance_matrix.size(0)

    # Calculate the normalized demand-density for each edge
    norm_demand_density = 2 * demands.view(n, 1) / (distance_matrix + 1e-6)  # Normalizing factor 2

    # Set penalties for edges exceeding capacity and scale the heuristics
    heuristics = norm_demand_density
    heuristics[torch.max(demands.view(n, 1), demands.view(1, n)) > 1] = -1
```

```
    return heuristics

# For CVRP500 and CVRP1000
def heuristics(distance_matrix: torch.Tensor, demands: torch.Tensor) -> torch.Tensor:
    excess_demand_penalty = torch.maximum(demands.sum() - demands, torch.tensor(0.))
    return 1 / (distance_matrix + 1e-6) - excess_demand_penalty
```

Heuristic 3: The best `ReEvo`-generated heuristics for CVRP_NCO_POMO.

```
# For CVRP200
def heuristics(distance_matrix: torch.Tensor, demands: torch.Tensor) -> torch.Tensor:
    total_demand = demands.sum()
    normalized_demand = demands / total_demand
    balanced_edge_weights = 1 / (distance_matrix + 1e-6)
    over_capacity_penalty = torch.clamp(demands.unsqueeze(1) + demands.unsqueeze(0) - 2, max=0)
    heuristics = balanced_edge_weights * normalized_demand.view(-1, 1) - normalized_demand - 2*over_capacity_penalty
    return heuristics

# For CVRP500
def heuristics(distance_matrix: torch.Tensor, demands: torch.Tensor) -> torch.Tensor:
    total_demand = torch.cumsum(demands, dim=0)
    vehicle_capacity = total_demand[-1]
    exceed_capacity_penalty = (total_demand.unsqueeze(1) > vehicle_capacity).float()
    unmet_demand_penalty = (vehicle_capacity - total_demand).clamp(min=0) / vehicle_capacity
    promisiness = (1 / (distance_matrix + 1)) * (1 - 0.5 * exceed_capacity_penalty - 0.5 * unmet_demand_penalty)
    return promisiness

# For CVRP1000
def heuristics(distance_matrix: torch.Tensor, demands: torch.Tensor) -> torch.Tensor:
    total_demand = demands.sum().item()
    demand_norm = demands / total_demand
    edge_savings = distance_matrix - demand_norm[:, None] - demand_norm
    return edge_savings
```

Heuristic 4: The best `ReEvo`-generated heuristics for CVRP_NCO_LEHD.

```
# Crossover
def crossover(parents: np.ndarray, n_pop: int) -> np.ndarray:
    n_parents, n_decap = parents.shape
    parents_idx = np.random.choice(n_parents, (n_pop, 2))
    crossover_points = np.random.randint(1, n_decap, n_pop)
    mask = np.tile(np.arange(n_decap), (n_pop, 1))
    offspring = np.where(mask < crossover_points.reshape(-1, 1),
                         parents[parents_idx[:, 0], :],
                         parents[parents_idx[:, 1], :])
    return offspring

# Mutation (A repairing step follows this mutation to ensure the feasibility of the population)
def mutation(population: np.ndarray, probe: int, prohibit: np.ndarray, size: int = 100) -> np.ndarray:
    p, n_decap = population.shape
    is_not_probe = np.all(population != probe, axis=1)
    is_not_prohibited = np.all(np.isin(population, prohibit, invert=True), axis=1)
    is_feasible = is_not_probe & is_not_prohibited
    mutation_mask = np.random.rand(p, n_decap) < 0.1
    mutation_values = np.random.randint(0, size, size=(p, n_decap))
    mutated_population = np.where(mutation_mask & is_feasible[:, None], mutation_values, population)
    return mutated_population
```

Heuristic 5: The best `ReEvo`-generated heuristic for DPP_GA.

```
def heuristics(distance_matrix: np.ndarray) -> np.ndarray:
    # Calculate the average distance for each node
    average_distance = np.mean(distance_matrix, axis=1)

    # Calculate the distance ranking for each node
    distance_ranking = np.argsort(distance_matrix, axis=1)

    # Calculate the mean of the closest distances for each node
    closest_mean_distance = np.mean(distance_matrix[np.arange(distance_matrix.shape[0])[:, None], distance_ranking[:, 1:5]],
        axis=1)

    # Initialize the indicator matrix and calculate ratio of distance to average distance
    indicators = distance_matrix / average_distance[:, np.newaxis]

    # Set diagonal elements to np.inf
    np.fill_diagonal(indicators, np.inf)

    # Adjust the indicator matrix using the statistical measure
    indicators += closest_mean_distance[:, np.newaxis] / np.sum(distance_matrix, axis=1)[:, np.newaxis]

    return indicators
```

Heuristic 7 presents the best heuristic found for TSP_ACO, which is generated when viewing TSP as a black-box COP. 'edge_attr' represents the distance matrix.

```python
import numpy as np
from sklearn.preprocessing import StandardScaler

def heuristics(edge_attr: np.ndarray) -> np.ndarray:
    num_edges = edge_attr.shape[0]
    num_attributes = edge_attr.shape[1]

    heuristic_values = np.zeros_like(edge_attr)

    # Apply feature engineering on edge attributes
    transformed_attr = np.log1p(np.abs(edge_attr))  # Taking logarithm of absolute value of attributes

    # Normalize edge attributes
    scaler = StandardScaler()
    edge_attr_norm = scaler.fit_transform(transformed_attr)

    # Calculate correlation coefficients
    correlation_matrix = np.corrcoef(edge_attr_norm.T)

    # Calculate heuristic value for each edge attribute
    for i in range(num_edges):
        for j in range(num_attributes):
            if edge_attr_norm[i][j] != 0:
                heuristic_values[i][j] = np.exp(-8 * edge_attr_norm[i][j] * correlation_matrix[j][j])

    return heuristic_values
```

Heuristic 7: The best ReEvo-generated heuristic for TSP_ACO.

```python
def heuristics(distance_matrix: np.ndarray, coordinates: np.ndarray, demands: np.ndarray, capacity: int) -> np.ndarray:
    num_nodes = distance_matrix.shape[0]

    # Calculate the inverse of the distance matrix
    inverse_distance_matrix = np.divide(1, distance_matrix, where=(distance_matrix != 0))

    # Calculate total demand and average demand
    total_demand = np.sum(demands)
    average_demand = total_demand / num_nodes

    # Calculate the distance from each node to the starting depot
    depot_distances = distance_matrix[:, 0]

    # Calculate the remaining capacity of the vehicle for each node
    remaining_capacity = capacity - demands

    # Initialize the heuristic matrix
    heuristic_matrix = np.zeros_like(distance_matrix)

    # Calculate the demand factor and distance factor
    demand_factor = demands / total_demand
    normalized_distance = distance_matrix / np.max(distance_matrix)
    distance_factor = depot_distances / (normalized_distance + np.finfo(float).eps)

    # Iterate over each node
    for i in range(num_nodes):

        # Calculate the heuristic value based on distance and capacity constraints
        heuristic_values = inverse_distance_matrix[i] * (1 / (normalized_distance[i] ** 2))

        # Adjust the heuristic values based on the remaining capacity
        heuristic_values = np.where(remaining_capacity >= demands[i], heuristic_values, 0)

        # Adjust the heuristic values based on the demand factor
        heuristic_values *= demand_factor[i] / average_demand

        # Adjust the heuristic values based on the distance factor
        heuristic_values *= distance_factor[i]
        heuristic_values[0] = 0  # Exclude the depot node

        # Adjust the heuristic values based on the capacity utilization
        utilization_factor = np.where(remaining_capacity >= demands[i], capacity - demands[i], 0)
        heuristic_values *= utilization_factor

        # Set the heuristic values for the current node in the heuristic matrix
        heuristic_matrix[i] = heuristic_values

    return heuristic_matrix
```

```python
def heuristics(prize: np.ndarray, distance: np.ndarray, maxlen: float) -> np.ndarray:
    n = prize.shape[0]
    heuristics = np.zeros((n, n))

    # Calculate the prize-to-distance ratio with a power transformation
    prize_distance_ratio = np.power(prize / distance, 3)

    # Find the indices of valid edges based on the distance constraint
    valid_edges = np.where(distance <= maxlen)

    # Assign the prize-to-distance ratio to the valid edges
    heuristics[valid_edges] = prize_distance_ratio[valid_edges]

    return heuristics
```

Heuristic 9: The best `ReEvo`-generated heuristic for OP_ACO.

Heuristic 10 presents the best heuristic found for MKP_ACO, which is generated when viewing MKP as a black-box COP. 'item_attr1' and 'item_attr2' represent the prizes and multi-dimensional weights of items, respectively.

```python
def heuristics(item_attr1: np.ndarray, item_attr2: np.ndarray) -> np.ndarray:
    n, m = item_attr2.shape

    # Normalize item_attr1 and item_attr2
    item_attr1_norm = (item_attr1 - np.min(item_attr1)) / (np.max(item_attr1) - np.min(item_attr1))
    item_attr2_norm = (item_attr2 - np.min(item_attr2)) / (np.max(item_attr2) - np.min(item_attr2))

    # Calculate the average value of normalized attribute 1
    avg_attr1 = np.mean(item_attr1_norm)

    # Calculate the maximum value of normalized attribute 2 for each item
    max_attr2 = np.max(item_attr2_norm, axis=1)

    # Calculate the sum of normalized attribute 2 for each item
    sum_attr2 = np.sum(item_attr2_norm, axis=1)

    # Calculate the standard deviation of normalized attribute 2 for each item
    std_attr2 = np.std(item_attr2_norm, axis=1)

    # Calculate the heuristics based on a combination of normalized attributes 1 and 2,
    # while considering the average, sum, and standard deviation of normalized attribute 2
    heuristics = (item_attr1_norm / max_attr2) * (item_attr1_norm / avg_attr1) * (item_attr1_norm / sum_attr2) * (1 /
        std_attr2)

    # Normalize the heuristics to a range of [0, 1]
    heuristics = (heuristics - np.min(heuristics)) / (np.max(heuristics) - np.min(heuristics))

    return heuristics
```

Heuristic 10: The best `ReEvo`-generated heuristic for MKP_ACO.

```python
def heuristics(demand: np.ndarray, capacity: int) -> np.ndarray:
    n = demand.shape[0]
    demand_normalized = demand / demand.max()

    same_bin_penalty = np.abs((capacity - demand[:, None] - demand) / capacity)
    overlap_penalty = (demand[:, None] + demand) / capacity

    heuristics = demand_normalized[:, None] + demand_normalized - same_bin_penalty - overlap_penalty

    threshold = np.percentile(heuristics, 90)
    heuristics[heuristics < threshold] = 0

    return heuristics
```

Heuristic 11: The best `ReEvo`-generated heuristic for BPP_ACO.

Heuristic 12 gives the best-`ReEvo` generated constructive heuristic for TSP. We used the best heuristic found in AEL [46] as the seed for `ReEvo`. As a result, our heuristic closely mirrors the one in AEL, scoring each node mostly using a weighted combination of the four factors.

```python
def select_next_node(current_node: int, destination_node: int, unvisited_nodes: set, distance_matrix: np.ndarray) -> int:
```

```
weights = {'distance_to_current': 0.4,
           'average_distance_to_unvisited': 0.25,
           'std_dev_distance_to_unvisited': 0.25,
           'distance_to_destination': 0.1}
scores = {}
for node in unvisited_nodes:
    future_distances = [distance_matrix[node, i] for i in unvisited_nodes if i != node]
    if future_distances:
        average_distance_to_unvisited = sum(future_distances) / len(future_distances)
        std_dev_distance_to_unvisited = (sum((x - average_distance_to_unvisited) ** 2 for x in future_distances) / len(
    future_distances)) ** 0.5
    else:
        average_distance_to_unvisited = std_dev_distance_to_unvisited = 0
    score = (weights['distance_to_current'] * distance_matrix[current_node, node] -
             weights['average_distance_to_unvisited'] * average_distance_to_unvisited +
             weights['std_dev_distance_to_unvisited'] * std_dev_distance_to_unvisited -
             weights['distance_to_destination'] * distance_matrix[destination_node, node])
    scores[node] = score
next_node = min(scores, key=scores.get)
return next_node
```

Heuristic 12: The best `ReEvo`-generated heuristic for TSP_constructive.

# F   Licenses for used assets

Table 11 lists the used assets and their licenses. Our code is licensed under the MIT License.

Table 11: Used assets and their licenses.

| Type | Asset | License | Usage |
|---|---|---|---|
| Code | NeuOpt [52] | MIT License | Evaluation |
| | GNNGLS [24] | MIT License | Evaluation |
| | EoH [47] | MIT License | Evaluation |
| | DeepACO [94] | MIT License | Evaluation |
| | DevFormer [31] | Apache-2.0 license | Evaluation |
| | POMO [37] | MIT License | Evaluation |
| | LEHD [51] | MIT License | Evaluation |
| Dataset | TSPLIB [67] | Available for any non-commercial use | Testing |
| | DPP PDNs [63] | Apache-2.0 license | Testing |

