# OpenReview forum: "ReEvo: Large Language Models as Hyper-Heuristics with Reflective Evolution"
_NeurIPS.cc/2024/Conference — NeurIPS 2024 poster_

### Official Review · Reviewer_2Tkz · 2024-07-07

**Soundness:** 3
**Presentation:** 3
**Contribution:** 2
**Rating:** 5
**Confidence:** 5

**Summary:**

This paper studies how to search for the best heuristics for combinatorial optimization problems (COPs) with large language models (LLM). The key idea is to use genetic programming (GP) to dynamically update the heuristics with LLM. Short-term reflection and long-term reflection are also incorporated in the GP algorithm as the guidance. The proposed methods are demonstrated on five different COPs and compared against other heuristic solvers and neural solvers.

**Strengths:**

1. Applying LLM to generate heuristics for COPs is novel.

2. The experiements are extensive and in detail.

3. The paper is well-written and the idea is easy to understand, except that the details of the GP algorithm are a bit vague.

**Weaknesses:**

1. The proposed ReEvo seems to rely on existing heuristic or neural solvers. Can ReEvo discover new heuristics independently? Incorporating LLM to generate heuristics for COPs is interesting, but it seems trivial if LLM can only be used as a post-processing to improve the existing methods.

2. The experiments are not conducted with state-of-the-art (SOTA) methods. For example, on TSP problem, can ReEvo improve the LKH-3 [1] solver? Can ReEvo improve the non-autoregressive neural solvers such as DIFUSCO [2], T2TCO [3]?

3. The writing in Sec. 4 needs improvement. For example, how the short-term and long-term reflection is utilized to improve heuristic design? how to guarantee the generated code is executable?

[1] Helsgaun, Keld. "An extension of the Lin-Kernighan-Helsgaun TSP solver for constrained traveling salesman and vehicle routing problems." Roskilde: Roskilde University 12 (2017): 966-980.

[2] Sun, Zhiqing, and Yiming Yang. "Difusco: Graph-based diffusion solvers for combinatorial optimization." Advances in Neural Information Processing Systems 36 (2023): 3706-3731.

[3] Li, Yang, et al. "From distribution learning in training to gradient search in testing for combinatorial optimization." Advances in Neural Information Processing Systems 36 (2024).

**Questions:**

I would appreciate the authors' response on my main concerns listed above.

**Limitations:**

I have no concerns regarding the impact of the manuscript.

---

> ### Author Rebuttal · Authors · 2024-08-05
>
> We deeply appreciate your time and effort in reviewing our work, and the insightful questions and suggestions you raised. We respond to your comments below.
>
>
> > W1: Reliance on existing heuristic or neural solvers.
>
> ReEvo can discover new heuristics independently but integration currently leads to better performance. In Section 5.4, we show that ReEvo can independently discover new constructive heuristics for TSP, which outperform the algorithm designed by traditional hyper-heuristics.
>
> ReEvo improves the key algorithmic components of existing methods, such as the perturbation in GLS, the crossover and mutation in GA for EDA, the heuristics in ACO, and the attention in NCO. We leave it to future work to further explore its potential to independently discover new complicated heuristics. Under an agentic framework, by first designing a skeleton of the program and then each individual function, we may be able to generate high-performing long programs using LLMs.
>
>
> > W2: Experiments with SOTA methods, such as T2TCO, DIFUSCO, and LKH-3.
>
> LHH is complementary to your suggested SOTA methods.
>
> - LKH-3 is a specialized solver for the TSP, which has been heavily optimized for the VRP over decades. LHHs, on the other hand, are more general and can be applied to any COP. They are especially suitable for cases where expert knowledge is lacking or the problem is treated as a black box.
>
> - DIFUSCO and T2TCO require optimal solutions for training, which may not be available for many problems. LHHs, on the other hand, only require heuristic evaluations. We compare against T2TCO on TSP500 and TSP1000, each with 128 test instances. Below we report the optimality gap and running time for solving all instances. They indicate the competitive performance of GLS-ReEvo.
>
> Method | TSP500 (gap; time) | TSP1000 (gap; time)
> --- | --- | ---
> T2TCO | 0.37% (16m) | 0.78% (55m)
> GLS (16 CPU cores) | 0.39% (12m) | 0.80 (2.6h)
> GLS-ReEvo (16 CPU cores) | 0.25% (12m) | 0.72% (2.6h)
>
> - LHH and general ML4CO [1] are complementary, each with its unique strengths and limitations. Some advantages of LHHs over ML4CO are:
>
>   - LHHs generate interpretable heuristics (code snippets), while ML4CO generates black-box parameterized policies. Interpretable heuristics offer insights for human designers and can be more reliable in practice when faced with dynamic environments, limited data, distribution shifts, or adversarial attacks.
>   - LHHs generate heuristics that are more efficient in terms of computational resources, as they do not require GPU during deployment.
>   - LHHs require only less than 100 heuristic evaluations and about 5 minutes to evolve a strong heuristic, while many ML4CO methods require millions of samples and days of training. When the solution evaluation is expensive, LHHs are more practical.
>   - LHHs only need some text-based (and even black-box) explanations to guide the search. ML4CO requires the development of NN architectures, hyperparameters, and training strategies, where informed inductive biases and manual tuning are crucial to guarantee performance.
>
>
> [1] https://github.com/Thinklab-SJTU/awesome-ml4co
>
>
> > W3-1: The writing in Sec. 4 needs improvement. For example, how the short-term and long-term reflection is utilized to improve heuristic design?
>
> We will improve the writing in Section 4 according to your suggestion.
>
> An example of both short- and long-term reflection is illustrated in Figure 1(b). The process prompts LLM to reflect upon the relative performance of two heuristics (short-term), and accumulate such knowledge over iterations (long-term). The inputs and outputs of the reflection process are all texts.
>
> **Short-term reflection.**
> - Inputs: Two heuristics to crossover and indicators of their relative performance.
> - Outputs: Text-based reflections upon the performance of the two heuristics, i.e. why the better heuristic is better, and how the worse heuristic can be improved. It is analogous to a verbal gradient derived from the performance comparison.
> - How it is utilized: ReEvo incorporates the short-term reflections into the crossover prompt, leading to more informed offspring.
>
> **Long-term reflection.**
> - Inputs: prior long-term reflections and short-term reflections of the current iteration.
> - Outputs: Text-based reflections upon the overall performance of the population, and strategic hints for better heuristic design.
> - How it is utilized: ReEvo incorporates the long-term reflections into the mutation prompt, leading to more strategic exploration.
>
>
> > W3-2: How to guarantee the generated code is executable?
>
> If the generated code is not executable, it is discarded.

---

> ### Author Response · Authors · 2024-08-13
> **Request for Feedback**
>
> As the author-reviewer discussion will end soon (< 24 hours from now), we would greatly appreciate it if you could take a moment to review our response. Please let us know if you have any further questions or concerns.

---

> > ### Comment · Reviewer_2Tkz · 2024-08-13
> >
> > Thank you for the rebuttal. While I have major concerns regarding your model's contributions to the CO community, you indeed conduct extensive experiments to demonstrate your model. So I will maintain my score, which is leaning acceptance.

---

> > > ### Author Response · Authors · 2024-08-13
> > > **Thanks for reviewing**
> > >
> > > Thank you again for reviewing and for your constructive feedback.

---

### Official Review · Reviewer_qdUQ · 2024-07-09

**Soundness:** 2
**Presentation:** 3
**Contribution:** 2
**Rating:** 5
**Confidence:** 3

**Summary:**

The paper presents a LLM-enhanced evolutionary algorithm to solve diverse combinatorial optimization problems. The method is evaludated on various COPs and heuristics, however, the advantage of proposed method compared to recent ML4CO methods that does not use LLM requires more specification.

**Strengths:**

1. The paper proposed a LLM-enhanced hyper-heuristic model to solve various combinatorial optimization problems.
2. Extensive experiments show the efficacy of the proposed method on five different COPs and multiple heuristics.

**Weaknesses:**

1. Some details of model is not fully specified from the text, including the meaning, output of "short-term reflection" and "long-term reflection". It is unclear how the process is done and how this process completes the "reflection"？
2. Though the proposed method has demonstrated the advances compared with heuristics, on specific problems (take TSP as example), there might be some recent models undiscussed in this paper that does not use heuristic but still achieve competitive results, such as [1,2] that conducts TSP experiments on 1000 and 10000.
3. The paper does not specify the costs of using LLM APIs, which could be real challenges when readers want to follow and reproduce the results in this paper.

[1] Qiu R, Sun Z, Yang Y. Dimes: A differentiable meta solver for combinatorial optimization problems[J]. Advances in Neural Information Processing Systems, 2022, 35: 25531-25546.
[2] Li Y, Guo J, Wang R, et al. From distribution learning in training to gradient search in testing for combinatorial optimization[J]. Advances in Neural Information Processing Systems, 2024, 36

**Questions:**

1. What result does figure 4(B) shows, the obejective value or the gap towards the optimal? Without specifying the evaluation metric, it is difficult to interpret the results.
2. How does the proposed method compared to recent ML4CO method on TSP [1,2], or probably other methods on VRP? From my experience, hyper-heuristic methods may not perform well as these methods on solution quality or solving time. What is the advantage of LLM-enhanced heuristics in these COPs?

**Limitations:**

The paper discuss some limitations, but the method may be limited by cost of LLM inference and problem scale.

---

> ### Author Rebuttal · Authors · 2024-08-05
>
> We deeply appreciate your time and effort in reviewing our work, as well as the insightful comments and questions you raised.
>
> > W1: Details of reflections.
>
> We will revise the writing for better clarity according to your suggestions. An example of both short- and long-term reflection is illustrated in Figure 1(b). The process prompts LLM to reflect upon the relative performance of two heuristics (short-term), and accumulate such knowledge over iterations (long-term). The inputs and outputs of the reflection process are all texts.
>
> **Short-term reflection.**
> - Inputs: Two heuristics to crossover and indicators of their relative performance.
> - Outputs: Text-based reflections upon the performance of the two heuristics, i.e. why the better heuristic is better, and how the worse heuristic can be improved. It is analogous to a verbal gradient derived from the performance comparison.
> - How it is utilized: ReEvo incorporates the short-term reflections into the crossover prompt, leading to more informed offspring.
> - Meaning: LLM reflects upon the performance of the two heuristics, providing insights for better heuristic design. It only targets the two heuristics for the current crossover, and does not consider the historical performance of the population.
>
> **Long-term reflection.**
> - Inputs: prior long-term reflections and short-term reflections of the current iteration.
> - Outputs: Text-based reflections upon the overall performance of the population, and strategic hints for better heuristic design.
> - How it is utilized: ReEvo incorporates the long-term reflections into the mutation prompt, leading to more strategic exploration.
> - Meaning: LLM accumulates knowledge over iterations, combining past experiences with new insights to offer long-term strategic hints.
>
> > W2: Undiscussed recent baselines.
>
> We comprehensively compare against 25 baselines across 6 COPs and 5 algorithmic types, under both black-box and white-box settings, and in terms of both LHH and generated heuristics. 8 of these baselines were published after 2023. We show that LHH can enhance ML4CO solvers in Section 5.5, involving TSP/CVRP with 1000 nodes.
>
> Here, we compare against the suggested SOTA baselines, on TSP500 and TSP1000 each with 128 test instances. Below we report the optimality gap and running time for solving all instances. They indicate the competitive performance of GLS-ReEvo. Please note that T2TCO requires optimal solutions for training, which may not be available for many problems, while ReEvo only requires heuristic evaluations.
>
> Method | TSP500 (gap; time) | TSP1000 (gap; time)
> --- | --- | ---
> DIMES | 1.76% (2.2h) | 2.46% (4.6h)
> T2TCO | 0.37% (16m) | 0.78% (55m)
> GLS (16 CPU cores) | 0.39% (12m) | 0.80 (2.6h)
> GLS-ReEvo (16 CPU cores) | 0.25% (12m) | 0.72% (2.6h)
>
>
> We respectfully bring to your attention that LHH and ML4CO [1] are complementary, each with its unique strengths and limitations. Some advantages of LHHs over ML4CO are:
>
> - LHHs generate interpretable heuristics (code snippets), while ML4CO usually generates black-box parameterized policies. Interpretable heuristics offer insights for human designers and can be more reliable in practice when faced with dynamic environments, limited data, distribution shifts, or adversarial attacks.
> - LHHs generate heuristics that are more efficient in terms of computational resources, as they do not require GPU during deployment.
> - LHHs require only less than 100 heuristic evaluations and about 5 minutes to evolve a strong heuristic, while many ML4CO methods require millions of samples and days of training. When the solution evaluation is expensive, LHHs are more practical.
> - LHHs only need some text-based (and even black-box) explanations to guide the search. ML4CO requires the development of NN architectures, hyperparameters, and training strategies, where informed inductive biases and manual tuning are crucial to guarantee performance.
>
> [1] https://github.com/Thinklab-SJTU/awesome-ml4co
>
> > W3: The cost of using LLM APIs
>
> As reported in Appendix B (line 858), each run costs about $0.06 when using GPT3.5 Turbo.
>
> > Q1: The y-axis of Figure 4(B).
>
> As indicated by the y-axis label, Figure 4(B) shows the objective value of the best generated heuristic, against the number of heuristic evaluations for an LHH.
>
> > Q2-1: Comparisons with recent ML4CO method on TSP and VRP.
>
> Please refer to our response to W2.
>
> > Q2-2: Advantage of LLM-enhanced heuristics over ML4CO methods.
>
> Please refer to our response to W2.
>
> > Potential limitations regarding LLM inference cost and problem scale
>
> Please also refer to our response to W2 and W3.
>
> **Inference cost.** Please note that we don't need LLM inference once the heuristic is generated, and the generated heuristics are efficient in terms of computational resources. For LHH search, deploying local LLMs is also feasible, as verified in our work with Llama 3.
>
> **Problem scale.** We have tested our method on large-scale problems, such as TSP1000 and CVRP1000. LHHs are currently most effective when integrated with existing algorithms. As long as the algorithm is scalable, its LHH-enhanced version is scalable as well.

---

> > ### Comment · Reviewer_qdUQ · 2024-08-13
> >
> > Thanks for the author's detailed response.  I agree with the point on the distinctive advantage of LHH compared with ML4CO, where LHH could benefit CO problem-solving. I would raise my score to 5.

---

> > > ### Author Response · Authors · 2024-08-13
> > > **Thanks for reviewing**
> > >
> > > Thank you again for reviewing and for your constructive feedback.

---

### Official Review · Reviewer_zrFg · 2024-07-11

**Soundness:** 3
**Presentation:** 3
**Contribution:** 1
**Rating:** 4
**Confidence:** 5

**Summary:**

This article proposes a large language model (LLM) assisted evolutionary computation (EC)-based method, to solve combinatorial optimization problems. It incorporates a reflection mechanism to enhance performance in black-box settings.

**Strengths:**

1. The method shows an impressive performance on several CO problems with a black-box setting.
2. The article is well-written.

**Weaknesses:**

1. The reflection technique has been well-developed and widely used in prompt engineering and code generation [1,2].
2. Evolution methods based on LLM have been adopted in EOH [3] and Funsearch [4], limiting the contribution to the framework.

[1] Reflexion: Language agents with verbal reinforcement learning. Advances in Neural Information Processing Systems 36 (2024).
[2] Promptbreeder: Self-referential self-improvement via prompt evolution. arxiv preprint arxiv:2309.16797 (2023).
[3] Evolution of Heuristics: Towards Efficient Automatic Algorithm Design Using Large Language Model. Forty-first International Conference on Machine Learning. 2024.
[4] Mathematical discoveries from program search with large language models. Nature 625.7995 (2024): 468-475.

**Questions:**

1. As an EC-based method, what is the primary distinction or improvement compared to EoH, FunSearch, and other similar methods [1,2,3]? The reflection technique was developed early and is widely used in prompt engineering and code generation(Shown in Weakness1). To what extent do you think designing more complex reflection and prompt engineering strategies contributes to optimization [4]?
2. In the ablation experiments, Please include ReEvo based on more open-source LLMs (such as DeepSeek and Gemini).
3. Testing on the online bin packing problem is recommended [5].
4. In Table 1, the result of EoH is drawn from the literature, testing it with the released code would be beneficial.
5. ReEvo requires three LLM calls to complete a single iteration, while other compared EC+LLM methods (such as EoH) only need one LLM call per iteration. Considering the comparison in the degree of LLM calls would be beneficial.

[1] Connecting large language models with evolutionary algorithms yields powerful prompt optimizers. arxiv preprint arxiv:2309.08532 (2023).
[2] Large Language Model-Aided Evolutionary Search for Constrained Multiobjective Optimization. arxiv preprint arxiv:2405.05767 (2024).
[3] Large language models as evolutionary optimizers." arxiv preprint arxiv:2310.19046 (2023).
[4] Are Large Language Models Good Prompt Optimizers?. arxiv preprint arxiv:2402.02101 (2024).
[5] Mathematical discoveries from program search with large language models. Nature 625.7995 (2024): 468-475.

---

> ### Author Rebuttal · Authors · 2024-08-05
>
> We deeply appreciate your time and effort in reviewing our work, as well as the insightful comments and questions you raised.
>
> > W1: The reflection technique has been well-developed and widely used in prompt engineering and code generation.
>
> Thank you for raising this point. Our work introduces a novel integration of reflection with evolutionary search. As noted by Reviewer v9WQ, this integration is effectively a novel technique to load in-context knowledge without incurring memory blowups.
>
> We respectfully draw your attention to the fact that the reflection techniques, which were initially proposed at NeurIPS 2023 [1, 2], are still under active development. Adapting reflection for a downstream application itself constitutes a valuable contribution; recent examples include its application to translation tasks [3, 4].
>
>
> > W2: Evolution methods based on LLM have been adopted in EoH and Funsearch, limiting the contribution to the framework.
>
> Thank you for highlighting the work done in EoH and FunSearch, which we extensively cite and acknowledge in our paper. These groundbreaking studies indeed lay the foundation for our work. We draw significant inspiration from them and aim to extend their contributions in meaningful ways.
>
> We believe that the combination of LLMs with EA for CO can be explored through three primary lenses: (1) the search algorithm, (2) the downstream CO applications (the search space), and (3) the evaluation methodologies (ways to evaluate how well we search). We believe our work contributes to all three aspects:
>
> - **Search Algorithm**: We introduce the Reflective Evolution method, demonstrating its superior sample efficiency.
> - **Applications**: Our work broadens the scope by applying this paradigm to five heterogeneous algorithmic types and six different combinatorial optimization problems, advancing the state of the art in GLS, EDA, ACO, and NCO methods.
> - **Evaluation Methodologies**: We employ fitness landscape analysis to explore the underlying mechanisms of our proposed method; we establish black-box experimental settings to ensure reliable comparisons and practical relevance to real-world applications.
>
> We believe that building upon and extending the foundations laid by EoH and FunSearch, as we have done, constitutes solid contributions.
>
>
> > Q1-1: The primary distinction or improvement over prior methods.
>
> Please refer to our response to W1 and W2.
>
> > Q1-2: The contributions of designing reflections and prompt engineering.
>
> We verify its contributions in Section 6. They are important for better sample efficiency (e.g. using less than 100 evaluations for designing strong heuristics). Sample efficiency is crucial for real-world applications where heuristic evaluation could be expensive.
>
> Reflection is more effective when using capable LLMs, such as GPT-3.5 Turbo and its successors, as discussed in prior works [1]. Allowing a large number of heuristic evaluations, or implementing weak open-source LLMs, could obscure the impact of reflection or other prompting techniques, as reported in [5].
>
> > Q2: Implement ReEvo based on more open-source LLMs for ablation.
>
> According to Shinn et al. [1], "self-reflection relies on the power of the LLM’s self-evaluation capabilities and not having a formal guarantee for success." Currently, many open-source LLMs are not capable enough to guarantee the statistically significant improvement of reflections. However, as LLM capabilities improve, we only expect this paradigm to get better over time [1].
>
> > Q3: Testing on the online bin packing problem.
>
> Below we present the comparative results on online BPP (weibull 5k), which are averaged over 3 runs.
>
> | EoH iteration         | 0      | 1      | 2      | 3      |
> | ----------------- | ------ | ------ | ------ | ------ |
> | total evaluations |        |        |        | 135    |
> | best obj. (mean)  | 4.2844 | 4.2844 | 4.2744 | 4.2511 |
> | best obj. (std)   | 0.0000 | 0.0000 | 0.0141 | 0.0286 |
> | best obj. (min)   | 4.2844 | 4.2844 | 4.2545 | 4.2145 |
>
> | ReEvo iteration | 1 | 2 | 3 | 4 | 5 | 6  | 7 | 8 | 9 | 10 | 11 |
> | --- | --- | --- | --- | --- | --- | --- | --- | --- | --- | --- | --- |
> | total evaluations | 30 | 40  | 45 | 55 | 60 | 70 | 75 | 85 | 90 | 100 | 105 |
> | best obj. (mean)  | 4.2844 | 4.2844 | 4.2844 | 4.2844 | 4.2844 | 3.1026 | 3.0560 | 3.0560 | 3.0560 | 1.6379 | 1.6379 |
> | best obj. (std)   | 0.0000 | 0.0000 | 0.0000 | 0.0000 | 0.0000 | 1.6713 | 1.6394 | 1.6394 | 1.6394 | 1.1943 | 1.1943 |
> | best obj. (min)   | 4.2844 | 4.2844 | 4.2844 | 4.2844 | 4.2844 | 0.7390 | 0.7390 | 0.7390 | 0.7390 | 0.7390 | 0.7390 |
>
> > Q4: Testing EoH with the released code
>
> We will test EoH with its released code and update the results.
>
> > Q5: Comparisons based on LLM calls.
>
> ReEvo doubles LLM calls when performing crossover. However, we believe LLM inference can be easily sped up by scaling up hardware, and is rapidly technically evolving on both algorithmic and hardware fronts. In contrast, evaluating the generated heuristics can be costly in real-world applications. Therefore, we believe that comparisons based on heuristic evaluations are more practical and meaningful.
>
> **References**:
>
> [1] Shinn et al., Reflexion: language agents with verbal reinforcement learning, NeurIPS 2023
>
> [2] Madaan et al., Self-Refine: Iterative Refinement with Self-Feedback, NeurIPS 2023
>
> [3] Wang et al., TASTE: Teaching Large Language Models to Translate through Self-Reflection, ACL 2024
>
> [4] Chen et al., DUAL-REFLECT: Enhancing Large Language Models for Reflective Translation through Dual Learning Feedback Mechanisms, ACL 2024
>
> [5] Zhang et al., Understanding the Importance of Evolutionary Search in Automated Heuristic Design with Large Language Models, 2024

---

> > ### Comment · Reviewer_zrFg · 2024-08-11
> >
> > Thank you for your explanations. I still have a few concerns that I would appreciate further clarification on.
> >
> > > W1 and W2 follow up:
> >
> > Firstly, while it is mentioned that LLM-based methods are less likely to cause memory blowups (such as compared to learning-based methods or complex heuristics like LKH3), this does not seem to highlight the specific contribution of integrating the self-reflection technique. Additionally, although I understand that reflection can enhance performance, it appears to me that the paradigm you proposed is more of a combination of the reflection technique and Evolutionary Computation (EC). This essentially allows the LLM to generate dynamic prompts to guide heuristic generation, as opposed to manually designed prompts used in the methods you compared.
> >
> > Moreover, in your experiments, did you control parameters such as the population size or LLM calls to prove that your proposed paradigm is overall superior to the state-of-the-art methods?
> >
> > > Q2 follow up:
> >
> > Could you clarify whether the performance of your proposed method varies significantly when applied to different LLMs? If so, should this be considered a major challenge for researchers attempting to follow your work, given that your method seems to rely heavily on the choice of LLM?
> >
> > > Q5 follow up:
> >
> > I still have some concerns about the logic presented. You mentioned that LLM inference can be accelerated through hardware scaling and that there are rapid advancements in both algorithmic and hardware technologies. However, currently, would it be fair to assume that comparisons based on LLM call counts could still lead to inconsistencies in terms of cost and performance?

---

> > > ### Author Response · Authors · 2024-08-12
> > > **Further clarifications**
> > >
> > > We appreciate your response. Below we clarify your remaining concerns.
> > >
> > > > LLM-based methods are less likely to cause memory blowups; mitigating memory blowups is not the contribution of EA-reflection integration.
> > >
> > > It seems there may be a misunderstanding regarding the concept of memory blowups in this context. Here, memory blowups do not refer to GPU memory usage (though that is also a concern). Instead, **it refers to the LLM agent's memory**. An LLM agentic architecture involves modules for (1) profile, (2) memory, (3) planning, and (4) action [1]. During its interactions with the environment (in our context, designing heuristics and receiving their fitness values as feedback), the agent's memory module stores the agent's experiences [1]. Although we want to load as much experience as possible into the LLM's context during inference, the context length is limited; the LLM's capability to process in-context knowledge/examples is also limited. This is the memory blowup we are referring to.
> > >
> > > In ReEvo, short-term reflections interpret the environmental feedback in each round of interaction. Long-term reflections distill the accumulated experiences and knowledge so that they can be loaded into context during inference without causing memory blowups. We hope this clarifies the contribution of EA-reflection integration in mitigating memory blowups.
> > >
> > > [1] A Survey on Large Language Model-based Autonomous Agents, 2024
> > >
> > > > Automated dynamic prompting for heuristic generation.
> > >
> > > Leveraging reflections to dynamically guide heuristic generation with accumulated knowledge is the core contribution of ReEvo. We hope ReEvo can inspire future research in combining EC with LLMs to eliminate manual prompt design and enhance the EoH performance with LLMs-derived prompts.
> > >
> > > > Control parameters for comparisons
> > >
> > > Due to the combinational nature of the parameter space, we believe it's reasonable to use the original parameters of the compared methods.
> > >
> > >
> > > > Does your method's performance vary significantly with different LLMs, posing a major challenge for researchers replicating it?
> > >
> > > LLM reasoning techniques generally show performance variations across different LLMs [1]. It is not challenging for researchers to follow our work, as it is evident that stronger LLMs are better at leveraging reflections or general reasoning techniques [1]. We recommend using advanced LLMs (e.g., GPT-3.5-turbo and its successors) to maximize the utility of reflection and other LLM techniques that require reasoning capabilities [1]. In the future, we only expect LLMs to become even more powerful and more capable of leveraging reflections [2].
> > >
> > > [1] Reasoning with Language Model Prompting: A Survey, 2024
> > >
> > > [2] Reflexion: Language agents with verbal reinforcement learning, 2023
> > >
> > >
> > > > Would it be fair to assume that comparisons based on LLM call counts could still lead to inconsistencies in terms of cost and performance?
> > >
> > >
> > > When a large number of heuristic evaluations are permitted, the performance of different EAs within the EoH/LHH context becomes nearly indistinguishable [1]. Therefore, we believe the primary goal of designing better EAs is to **address scenarios where heuristic evaluations are costly**, which is common in many industrial applications. To this end, we propose ReEvo and adopt an experimental setting that limits heuristic evaluations.
> > >
> > > LLM inference today, whether through local models or commercial APIs, **is already highly cost-effective**—both in terms of expense (e.g., about $0.0003 per call in ReEvo using GPT-3.5-turbo) and time (e.g., less than one second on average with asynchronous API calls or batched inference). These costs are negligible compared to real-world heuristic evaluations, which, even on simplified academic benchmarks, can take over 20 minutes per evaluation in EDA problems.
> > >
> > > **LLM reasoning techniques** (e.g., Chain of Thought or CoT) offer another perspective for fair comparisons. Although variants like Tree of Thought (ToT), Graph of Thought (GoT), and MCTS-based methods require more LLM calls, performance is typically assessed by the final evaluation outcomes, such as k-shot solution assessments, similar to k heuristic evaluations in the EoH/LHH context.
> > >
> > > If one has to compare EoH/LHHs based on LLM inference costs, it is essential to consider token usage, as tokens determine both the cost and time, not the number of LLM calls. Requiring additional heuristic descriptions, as in EoH, increases these costs, which ReEvo avoids.
> > >
> > > In summary, comparing LLMs based on the number of LLM calls is less reasonable or meaningful.
> > >
> > >
> > > [1] Understanding the Importance of Evolutionary Search in Automated Heuristic Design with Large Language Models, 2024

---

> > > > ### Author Response · Authors · 2024-08-13
> > > > **Request for Feedback**
> > > >
> > > > As the author-reviewer discussion will end soon (< 24 hours from now), we would greatly appreciate it if you could take a moment to review our clarifications. Please let us know if you have any further questions or concerns.

---

> > > > > ### Comment · Reviewer_zrFg · 2024-08-14
> > > > >
> > > > > Thank you for your clarification.
> > > > >
> > > > > 1. I still have concerns about the performance under different LLMs. I believe it is necessary to test the performance under different LLMs:
> > > > >
> > > > > 1.) GPT is one of the LLMs with the best overall performance; however, it might not necessarily be the best model for heuristic design combined with search methods. For instance, certain LLMs may provide better results in tasks. Additionally, it is worth exploring whether different LLMs are better suited for reflection and code generation tasks, as well as choosing the most appropriate model for this specific task.
> > > > >
> > > > > 2.) Testing different LLMs would help in understanding the performance boundaries of the method and assist users in various application scenarios in selecting the most suitable LLM configuration. The most optimal LLM configuration for different combinatorial optimization heuristic design tasks may vary.
> > > > >
> > > > > 3.) GPT is closed-source and cannot be deployed locally, which makes it challenging to use in many industry scenarios where API is unavailable.
> > > > >
> > > > > 2. The results on bin packing are not convincing. With a specific setting, the results are much worse than those in the FunSearch and EoH papers.
> > > > >
> > > > > I know it's difficult to add experiments during the rebuttal period. I will increase my score by 1. But I still think it's an incremental work based on FunSearch and EoH with marginal novelty.

---

> > > > > > ### Author Response · Authors · 2024-08-14
> > > > > > **Thanks for reviewing**
> > > > > >
> > > > > > Thank you for reviewing. In the future update, we commit to adding more experiments based on your suggestions.
> > > > > >
> > > > > > We would like to, again, respectfully argue that our work meaningfully extends FunSearch and EoH through three primary lenses: (1) the search algorithm, (2) the downstream CO applications, and (3) the evaluation methodologies.
> > > > > >
> > > > > > - Search Algorithm: We introduce the Reflective Evolution, demonstrating its superior sample efficiency.
> > > > > > - Applications: Our work broadens the scope by applying this paradigm to five heterogeneous algorithmic types and six different combinatorial optimization problems, advancing the state of the art in GLS, EDA, ACO, and NCO methods.
> > > > > > - Evaluation Methodologies: We employ fitness landscape analysis to explore the underlying mechanisms of our proposed method; we establish black-box experimental settings to ensure reliable comparisons and practical relevance to real-world applications.

---

### Official Review · Reviewer_v9WQ · 2024-07-12

**Soundness:** 3
**Presentation:** 4
**Contribution:** 3
**Rating:** 7
**Confidence:** 4

**Summary:**

The paper's contributions consist of multiple sections, which could be orthogonal:
1. The paper proposes a new ReEvo algorithm within the class of string-mutation evolutionary search methods. The core idea is to add a "Reflection LLM" which observes patterns over the history of trials, and proposes a new generation instruction for the regular Generator LLM.
2. This algorithm is applied on the area of combinatorial optimization, to mutate the code specifying a search heuristic.

Experiments are conducted to:
1. Show better heuristic search over a variety problems and situations like Travelling Salesman (TSP), Ant Colony Optimization, Electronic Design Automation, and Neural Combinatorial Optimization
2. Ablate the importance of using the "Reflection LLM" especially over longer terms, and outperformance over other string-mutation methods like EoH.

**Strengths:**

* This paper is very well written and structured. It demonstrates a strong understanding of the current literature and the nuanced differences against previous works, and their strengths/weaknesses.
* The paper supports itself well with numerous experiments among multiple dimensions (e.g. varying combinatorial optimization applications, ablation studies on its own components, comparisons against other string-mutation baselines).
* Overall, the paper does not have any glaring weaknesses and should be a solid accept.

**Weaknesses:**

* The current writing style is a bit too strong on the evolutionary side (e.g. targeted towards an audience like GECCO), which makes parts of the paper unmotivated to readers who are not deep into the evolutionary literature. It becomes too easy to view the paper as a incremental improvement in the class of evolutionary string-mutation methods, which in itself is a good contribution, but not groundbreaking given previous works (PromptBreeder, FunSearch, and other works that the paper itself has cited).
  * There are possibly better ways to phrase the paper to make the method more natural. For example, I believe the Reflection LLM with its short-term + long-term variants, is effectively a technique to load in-context samples without incurring memory-blowups.
* It is unclear whether ReEvo can be applied to any other string-based optimization scenarios (e.g. Prompt optimization, generic code search). Currently the Reflection LLM is very targeted towards summarizing combinatorial heuristics. If so, this would be nice to touch upon, or if not, this should be listed in the limitations section.
  * Following up on the above comment, this could imply that ReEvo is over-engineered at the moment; is there a more general and simpler variant of it, applicable to any string problems?

**Questions:**

Please address my weaknesses above.

**Limitations:**

Yes, in Section 7 (Conclusion).

---

> ### Author Rebuttal · Authors · 2024-08-05
>
> We deeply appreciate your time and effort in reviewing our work, the insightful questions and suggestions you raised, and your recognition of our contributions. We respond to your comments below.
>
> > W1: The paper's evolutionary focus may be limiting its appeal and perceived novelty to a broader audience, and rephrasing it to emphasize the method's natural approach could be beneficial.
>
> Thank you for providing a fresh perspective on the technical presentation. We will adopt your suggestion to generalize the description of the reflection technique. As you suggest, this approach loads in-context knowledge without causing memory blowups. We believe this could broaden its appeal to a wider audience.
>
>
> > W2: The paper should clarify ReEvo's applicability to other string-based optimization scenarios and consider discussing a more generalized version of the method.
>
> Thank you for this valuable suggestion. ReEvo is generally applicable to other string-based optimization scenarios as long as the reflecting over the relative performance of strings is meaningful. We exemplify its generality by applying it to prompt learning. We evolve prompts for formal logic reasoning task from MMLU, using random search, vanilla GA, and ReEvo, respectively. The results below present the classification accuracy against the number of prompt evaluations, where ReEvo demonstrates superior sample efficiency.
>
> \# prompt evaluations | 4 | 8 | 12 | 16 | 20
> --- | --- | --- | --- | --- | ---
> Random Search | 10  $\pm$ 0 % | 10  $\pm$ 0 % | 10  $\pm$ 0 % | 10  $\pm$ 0 % | 10  $\pm$ 0 %
> Vanilla GA | 10  $\pm$ 0 % | 22 $\pm$ 8 %| 25 $\pm$ 12 %| 37 $\pm$ 8 %| 38 $\pm$ 9%
> ReEvo | 10  $\pm$ 0 % | 37 $\pm$ 5 %| 43 $\pm$ 2 %| 45 $\pm$ 0 %| 45 $\pm$ 0%

---

> > ### Comment · Reviewer_v9WQ · 2024-08-07
> > **A different writing strategy may have been better.**
> >
> > Thanks for the clarification. As I mentioned, because the focus of the paper is on combinatorial heuristics, this is causing the other reviewers to require multiple comparisons to SOTA combinatorial optimizers, which I'm not sure is even worth the time to address.
> >
> > I think the strategy of phrasing the paper's contributions in terms of the general algorithm (rather than specific combinatorial application) may have been better.
> >
> > I retain my score, but acknowledge the above issue may lead to a possible rejection of the paper.

---

> > > ### Author Response · Authors · 2024-08-08
> > > **Thank you for reviewing**
> > >
> > > Thank you for reviewing. We will follow your suggestions to rephrase the paper's contributions.
> > >
> > > Regarding the comparison with SOTA heuristic solvers (e.g. LKH), LHHs are general and versatile. They are especially suitable for cases where expert knowledge is lacking or the problem is treated as a black box.
> > >
> > > Regarding the comparison with SOTA ML4CO optimizers, LHHs demonstrate unique strengths:
> > >
> > > - LHHs generate interpretable heuristics (code snippets), while ML4CO usually generates black-box parameterized policies. Interpretable heuristics offer insights for human designers and can be more reliable in practice when faced with dynamic environments, limited data, distribution shifts, or adversarial attacks.
> > >
> > > - LHHs generate heuristics that are more efficient in terms of computational resources, as they do not require GPU during deployment.
> > >
> > > - LHHs require only less than 100 heuristic evaluations and about 5 minutes to evolve a strong heuristic, while many ML4CO methods require millions of samples and days of training. When the solution evaluation is expensive, LHHs are more practical.
> > >
> > > - LHHs only need some text-based (and even black-box) explanations to guide the search. ML4CO requires the development of NN architectures, hyperparameters, and training strategies, where informed inductive biases and manual tuning are crucial to guarantee performance.

---

### Author Response · Authors · 2024-08-13
**Request for Feedback**

Dear reviewers, as the author-reviewer discussion phase is coming to an end, please let us know if there are further questions or concerns to address. Thanks in advance for your time and consideration!

---

### Decision · Program_Chairs · 2024-09-25

**Decision:**

Accept (poster)

**Comment:**

The paper presents a novel and general framework that uses LLMs to evolve hyper-heuristics. The framework is built primarily from existing LLM-based components, but coheres well and achieves its goal of an approach that feels like what it's like to design heuristics as a human, leading to interpretable insights in specific results, and allowing the reader to reflect on their own scientific/engineering process through this exercise in formalizing it. The method should be inspiring for many long-term evolutionary improvement approaches.

The main remaining concern about the paper is its similarity to a very recent work EoH, which also frames itself in the lineage of hyper-heuristics. The present work is sufficiently novel to merit its own publication, and the empirical results show strong benefits compared with EoH. However, I suggest the authors clearly state that EoH was developed _concurrently_ to this work, and enumerate the exact differences. To further clarify the paper's place in the literature, it would also be helpful to reference the earliest work using LLMs for mutations [1] and crossover [2] of code.

Beyond incorporating the changes agreed upon in the discussion with reviewers, I would also encourage a longer dedicated limitations section; since the method appears so general, it would be helpful to clearly identify scenarios where it is not expected to be a good choice.

[1] Lehman et al. "Evolution through Large Models" arXiv:2206.08896 (2023, Handbook of Evolutionary Machine Learning)
[2] Meyerson et al. "Language Model Crossover: Variation through Few-Shot Prompting" arXiv:2302.12170